# EXPRESSIVITY OF ReLU-NETWORKS UNDER CONVEX RELAXATIONS

**Maximilian Baader,**[*] **Mark Niklas Müller,**[*] **Yuhao Mao & Martin Vechev**
Department of Computer Science
ETH Zurich, Switzerland
{mbaader,mark.mueller,yuhao.mao,martin.vechev}@inf.ethz.ch

## ABSTRACT

Convex relaxations are a key component of training and certifying provably safe neural networks. However, despite substantial progress, a wide and poorly understood accuracy gap to standard networks remains, raising the question of whether this is due to fundamental limitations of convex relaxations. Initial work focused on the simple and widely used IBP relaxation. It revealed that some univariate, convex, continuous piecewise linear (CPWL) functions cannot be encoded by any ReLU network such that its IBP-analysis is precise. To explore whether this limitation is shared by more advanced convex relaxations, we conduct the first in-depth study on the expressive power of ReLU networks across all commonly used convex relaxations. We show that: (i) more advanced relaxations allow a larger class of *univariate* functions to be expressed as precisely analyzable ReLU networks, (ii) more precise relaxations can allow exponentially larger solution spaces of ReLU networks encoding the same functions, and (iii) even using the most precise single-neuron relaxations, it is impossible to construct precisely analyzable ReLU networks that express *multivariate*, convex, monotone CPWL functions.

## 1 INTRODUCTION

With the increased deployment of neural networks in mission-critical applications, formal robustness guarantees against adversarial examples (Biggio et al., 2013; Szegedy et al., 2014) have become an important and active field of research. Many popular certification methods (Zhang et al., 2018; Singh et al., 2018; 2019a;b) provide such safety guarantees by using convex relaxations to compute over-approximations of a network's reachable set w.r.t. an adversary specification. However, despite significant progress, a wide and poorly understood accuracy gap between robust and conventional networks remains. This raises the fundamental question:

Table 1: Expressivity of different relaxations. Novel results are red ✗ and green ✓. Previous results are in black (✗, ?). M: monotone, C: convex, MC: monotone and convex.

| $\mathcal{X}$ | Relaxation | CPWL | M-CPWL | C-CPWL | MC-CPWL |
|---|---|---|---|---|---|
| $\mathbb{R}$ | IBP | ✗ | ✓ | ✗ | ✓ |
| | DEEPPOLY-0 | ? | ✓ | ✓ | ✓ |
| | DEEPPOLY-1 | ? | ✓ | ✓ | ✓ |
| | $\Delta$ | ? | ✓ | ✓ | ✓ |
| | Multi-Neuron$_\infty$ | ✓ | ✓ | ✓ | ✓ |
| $\mathbb{R}^d$ | $\Delta$ | ✗ | ✗ | ✗ | ✗ |

*Is the expressivity of ReLU-networks under convex relaxations fundamentally limited?*

Investigating this question, Mirman et al. (2022) prove that, the class of convex, continuous-piecewise-linear (CPWL) functions *cannot* be encoded as ReLU-networks such that their analysis with the simple IBP-relaxation (Gehr et al., 2018; Gowal et al., 2018), is *precise*.

**This Work: Expressivity of Common Relaxations** To investigate whether this limitation of IBP is fundamental to all single-neuron convex relaxations, we conduct the first in-depth study on the expressive power of ReLU networks under *all* commonly used relaxations. To this end, we consider CPWL functions, naturally represented by ReLU networks, and two common restrictions, convexity and monotonicity. We illustrate key findings in Table 1, showing novel results as red ✗ and green ✓.

---

[*]Equal contribution.

**Key Results on Univariate Functions**    In this work, we prove the following key results:

- The most precise single-neuron relaxation, $\Delta$ (Wong & Kolter (2018)), and the popular DEEPPOLY-relaxation (Singh et al., 2019b; Zhang et al., 2018) do not share IBP's limitation and can express univariate, *convex*, CPWL functions precisely.
- All considered relaxations, including IBP, can express univariate, *monotone*, CPWL functions precisely.
- The $\Delta$-relaxation permits an exponentially larger network solution space for convex CPWL functions compared to the less precise DEEPPOLY-relaxation.
- Multi-neuron relaxations (Singh et al., 2019a; Müller et al., 2022) can express all univariate, CPWL functions precisely using a single layer.

Having thus shown that, for *univariate functions*, the expressivity of ReLU networks under convex relaxations *is not fundamentally limited*, we turn our analysis to multivariate functions.

**Key Results on Multivariate Functions**    In this setting, we prove the following result:

- No single-neuron convex relaxation can precisely express even the heavily restricted class of multivariate, convex, monotone, CPWL functions.

Interestingly, the exact analysis of such monotone functions on box input regions is trivial, making the failure of convex relaxations even more surprising. In fact, CPWL functions as simple as $f(x, y) = \max(x, y) = y + \mathrm{ReLU}(x - y)$ cannot be encoded by any finite ReLU network such that its $\Delta$-analysis is precise. We thus conclude that, for *multivariate functions*, the expressivity of ReLU networks under single-neuron convex relaxations *is fundamentally limited*.

**Implications of our Results for Certified Training** While we believe our results to be of general interest, they have particularly interesting implications for certified training. In this area, all state-of-the-art methods (Müller et al., 2023; Mao et al., 2023; Palma et al., 2023) are based on the simple IBP-relaxation even though it induces strong regularisation which severely reduces accuracy. While Jovanović et al. (2022) show that more precise relaxations induce significantly harder optimization problems, it remains an open question whether solving these would actually yield networks with better performance. Our results represent a major step towards answering this question.

Specifically in the univariate setting, we show that more precise relaxations increase expressivity (see Table 1) and lead to larger network solution spaces (compare Theorems 11 and 15). Thus, we hypothesize that using them during training yields a larger effective hypothesis space for the same network architecture. Importantly, this implies that networks with higher performance could indeed be obtained if we can overcome the optimization issues described by Jovanović et al. (2022).

However, in the multivariate setting, perhaps surprisingly, we show that even the most precise single-neuron relaxations severely limit expressivity (see Corollary 21). This highlights the need for further study of more precise analysis methods such as multi-neuron or non-convex relaxations.

## 2    BACKGROUND ON CONVEX RELAXATIONS

**Notation**    We denote vectors with bold lower-case letters $\boldsymbol{a} \in \mathbb{R}^n$, matrices with bold upper-case letters $\boldsymbol{A} \in \mathbb{R}^{n \times d}$, and sets with upper-case calligraphic letters $\mathcal{A} \subset \mathbb{R}$. Inequalities $\boldsymbol{a} \geq \boldsymbol{b}$ between vectors are elementwise. We refer to a hyperrectangle $\mathcal{B} \subset \mathbb{R}^n$ as a box. Further, we consider general (finite) ReLU networks $h$ with arbitrary skip connections, including CNNs and ResNets.

### 2.1    CONVEX RELAXATIONS IN NEURAL NETWORK CERTIFICATION

We call a classifier $H \colon \mathcal{X} \to \mathcal{Y}$ locally robust around an input $\boldsymbol{x} \in \mathcal{X}$ if it predicts the same, correct class $y \in \mathcal{Y}$ for all similar inputs $\mathcal{B}_p^\epsilon(x) \coloneqq \{\boldsymbol{x}' \in \mathcal{X} \mid \|\boldsymbol{x} - \boldsymbol{x}'\|_p \leq \epsilon\}$. To prove that a classifier is locally robust, we thus have to show that $H(x') = H(x) = y, \forall x' \in \mathcal{B}$. For a neural network predicting $H(x) \coloneqq \arg\max_i h(x)_i$, this is equivalent to showing that the logit of the target class is always greater than that of all other classes, i.e., $0 < \min_{x' \in \mathcal{B}, i \neq y} h(x')_y - h(x')_i$. As solving this non-convex optimization problem exactly is generally NP-complete (Katz et al., 2017), state-of-the-art neural network verifiers (Brix et al., 2023) relax it to an efficiently solvable convex optimization

problem. To this end, we replace the non-linear activation functions with convex relaxations in their input-output space, allowing us to compute linear bounds on the output $h(\boldsymbol{x})$:

$$\{\boldsymbol{A}_{l_i}\boldsymbol{x} + b_{l_i}\}_{i\in\mathcal{L}} \leq h(\boldsymbol{x}) \leq \{\boldsymbol{A}_{u_j}\boldsymbol{x} + b_{u_j}\}_{j\in\mathcal{U}},$$

for some input region $\mathcal{B}_p^\epsilon(x)$, with index sets $\mathcal{L}$ and $\mathcal{U}$. These bounds can in-turn be bounded by $\boldsymbol{l}_y = \min_{\boldsymbol{x}\in\mathcal{B}}\max_{i\in\mathcal{L}}(\boldsymbol{A}_{l_i}\boldsymbol{x} + b_{l_i}) \in \mathbb{R}$ and $\boldsymbol{u}_y$ analogously. Hence, we have $\boldsymbol{l}_y \leq \boldsymbol{h}(\boldsymbol{x}) \leq \boldsymbol{u}_y$.

**IBP** Interval bound propagation (Mirman et al., 2018; Gehr et al., 2018; Gowal et al., 2018) only considers elementwise, constant bounds of the form $\boldsymbol{l} \leq \boldsymbol{v} \leq \boldsymbol{u}$. Affine layers $\boldsymbol{y} = \boldsymbol{W}\boldsymbol{v} + \boldsymbol{b}$ are thus also relaxed as

$\frac{\boldsymbol{W}(\boldsymbol{l}+\boldsymbol{u})-|\boldsymbol{W}|(\boldsymbol{u}-\boldsymbol{l})}{2} + \boldsymbol{b} \leq \boldsymbol{W}\boldsymbol{v} + \boldsymbol{b} \leq \frac{\boldsymbol{W}(\boldsymbol{l}+\boldsymbol{u})+|\boldsymbol{W}|(\boldsymbol{u}-\boldsymbol{l})}{2} + \boldsymbol{b}$,

where $|\cdot|$ the elementwise absolute value. ReLU functions are relaxed by their concrete lower and upper bounds $\mathrm{ReLU}(\boldsymbol{l}) \leq \mathrm{ReLU}(\boldsymbol{v}) \leq \mathrm{ReLU}(\boldsymbol{u})$, illustrated in Fig. 1.

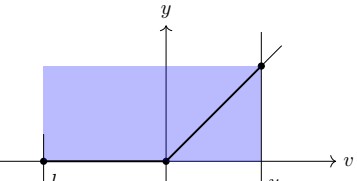

Figure 1: IBP-relaxation of a ReLU with bounded inputs $v \in [l, u]$.

**DeepPoly (DP)** DeepPoly, introduced by Singh et al. (2019b), is mathematically identical to CROWN (Zhang et al., 2018) and based on recursively deriving linear bounds of the form

$$\boldsymbol{A}_l\boldsymbol{x} + \boldsymbol{a}_l \leq \boldsymbol{v} \leq \boldsymbol{A}_u\boldsymbol{x} + \boldsymbol{a}_u$$

on the outputs of every layer. While this allows affine layers to be handled exactly, ReLU layers $\boldsymbol{y} = \mathrm{ReLU}(\boldsymbol{v})$ are relaxed neuron-wise, using one of the two relaxations illustrated in Fig. 2

$$\boldsymbol{\lambda}\boldsymbol{v} \leq \mathrm{ReLU}(\boldsymbol{v}) \leq (\boldsymbol{v} - \boldsymbol{l})\frac{\boldsymbol{u}}{\boldsymbol{u} - \boldsymbol{l}},$$

where product and division are elementwise. Typically, the lower-bound slope $\lambda \in \{0, 1\}$ is chosen depending on the input bounds $l$ and $u$. In this work, however, we analyze the relaxations obtained by always choosing the same lower-bound, which we denote with DEEPPOLY-0 (DP-0, green in Fig. 2) and DEEPPOLY-1 (DP-1, blue).

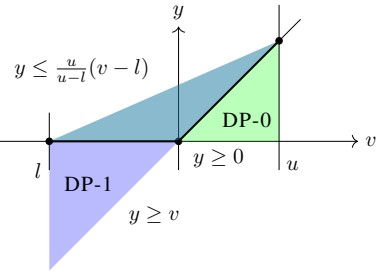

Figure 2: DEEPPOLY-1 (blue) and DEEPPOLY-0 (green) ReLU abstraction with bounded inputs $v \in [l, u]$.

**Triangle-Relaxation ($\Delta$)** In contrast to the above convex relaxations, the $\Delta$-relaxation (Wong & Kolter, 2018; Dvijotham et al., 2018; Salman et al., 2019; Qin et al., 2019) maintains multiple linear upper- and lower-bounds on every network activation $v$. We write

$$\left.\begin{array}{c}\boldsymbol{A}_{l_1}\boldsymbol{x} + \boldsymbol{a}_{l_1}, \\ \vdots \\ \boldsymbol{A}_{l_n}\boldsymbol{x} + \boldsymbol{a}_{l_n},\end{array}\right\} \leq \boldsymbol{v} \leq \left\{\begin{array}{c}\boldsymbol{A}_{u_1}\boldsymbol{x} + \boldsymbol{a}_{u_1}, \\ \vdots \\ \boldsymbol{A}_{u_n}\boldsymbol{x} + \boldsymbol{a}_{u_n}.\end{array}\right.$$

Unstable ReLU activation $\boldsymbol{y} = \mathrm{ReLU}(\boldsymbol{v})$ with $\boldsymbol{l} < \boldsymbol{0} < \boldsymbol{u}$ are relaxed with their convex hull as illustrated in Fig. 3

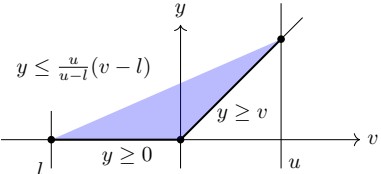

Figure 3: $\Delta$-relaxation of a ReLU with bounded inputs $v \in [l, u]$.

$$\left.\begin{array}{c}\boldsymbol{0} \\ \boldsymbol{v}\end{array}\right\} \leq \mathrm{ReLU}(\boldsymbol{v}) \leq (\boldsymbol{v} - \boldsymbol{l})\frac{\boldsymbol{u}}{\boldsymbol{u} - \boldsymbol{l}},$$

where again, product and division are elementwise. Note that this can lead to an exponential growth (with the depth of the network) in the number of constraints for any given activation.

**Multi-Neuron Relaxation (MN)** All methods introduced so far relax activation functions neuron-wise and are thus limited in precision by the (single neuron) convex relaxation barrier (Salman et al., 2019), i.e., the activation function's convex hull in their input-output space.

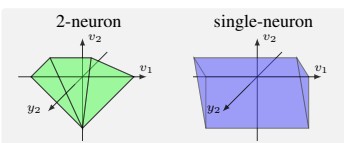

Figure 4: Comparison of a 2-neuron (green) and single-neuron (blue) relaxation projected into $y_2$-$v_1$-$v_2$-space for ReLU activations $y_i = \mathrm{ReLU}(v_i)$.

Multi-neuron relaxations, in contrast, compute the convex hull in the joint input-output space of multiple neurons in the same layer (Singh et al., 2019a; Müller et al., 2022), or consider multiple inputs jointly (Tjandraatmadja et al., 2020). We illustrate the increase in tightness in Fig. 4 for a group of just $k = 2$ neurons.

## 2.2 Definitions

We now define the most important concepts for this work.

**Definition 1** (CPWL). We denote the set of continuous piecewise linear functions $f \colon \mathcal{X} \to \mathcal{Y}$ by $\mathrm{CPWL}(\mathcal{X}, \mathcal{Y})$. Further, if $\mathcal{X}$ is some interval $\mathbb{I} \subset \mathbb{R}$, then we enumerate the points where $f$ changes slope and call them $x_i$, where $0 \leq i \leq n$, $i < j$ implies $x_i < x_j$, and $\mathcal{X} = [x_0, x_n]$.

All CPWL functions $f \colon \mathbb{I} \to \mathbb{R}$ satisfy $f(x) = f(x_i) + (x - x_i)\frac{f(x_{i+1}) - f(x_i)}{x_{i+1} - x_i}$ for $x \in [x_i, x_{i+1}]$. We denote by M-CPWL, C-CPWL, and MC-CPWL the class of monotone (M), convex (C), and monotone & convex (MC) CPWL functions, respectively. We say a network $h$ encodes a function $f$ if and only if they are equal on all inputs $\boldsymbol{x} \in \mathcal{X}$:

**Definition 2** (Encoding). A neural network $h \colon \mathcal{X} \to \mathcal{Y}$ *encodes* a function $f \colon \mathcal{X} \to \mathcal{Y}$ if and only if for all $x \in \mathcal{X}$ we have $h(x) = f(x)$.

In the following, $D$ denotes a convex relaxation and can be IBP, DeepPoly-0 (DP-0), DeepPoly-1 (DP-1), $\Delta$, or Multi-Neuron (MN). We now call the over-approximation of a network's graph (set of input-output tuples) obtained with domain $D$ its $D$-analysis:

**Definition 3** (Analysis). Let $h \colon \mathcal{X} \to \mathcal{Y}$ be a network, $D$ a convex relaxation, and $\mathcal{B} \subset \mathcal{X}$ an input box. We denote by $h^D(\mathcal{B})$ the polytope in $h$'s input-output space containing the graph $\{(\boldsymbol{x}, h(\boldsymbol{x})) \mid \boldsymbol{x} \in \mathcal{B}\} \subseteq h^D(\mathcal{B}) \subseteq \mathcal{X} \times \mathcal{Y}$ of $h$ on $\mathcal{B}$, as obtained with $D$ and refer to it as the $D$-*analysis* of $h$ on $\mathcal{B}$.

For $\mathcal{Y} \subseteq \mathbb{R}$, we denote the interval bounds of $f$ on $\mathcal{B}$ by $[\underline{f(\mathcal{B})}, \overline{f(\mathcal{B})}] := [\min_{\boldsymbol{x} \in \mathcal{B}} f(\boldsymbol{x}), \max_{\boldsymbol{x} \in \mathcal{B}} f(\boldsymbol{x})]$ and, similarly, the interval bounds implied by $h^D(\mathcal{B})$ as $[\underline{h^D(\mathcal{B})}, \overline{h^D(\mathcal{B})}] := [\min_{(\boldsymbol{x}, y) \in h^D(\mathcal{B})} y, \max_{(\boldsymbol{x}, y) \in h^D(\mathcal{B})} y]$.

As any $D$-analysis of $h$ captures the set of all possible outputs $h(\boldsymbol{x}), \boldsymbol{x} \in \mathcal{B}$, it is of key interest to us to investigate when the analysis does not lose precision. Specifically, whether the linear output bounds $h^D(\mathcal{B})$ do not exceed the interval bounds of $f$ on $\mathcal{B}$ anywhere on $\mathcal{B}$:

**Definition 4** (Precise). Let $h$ be a network encoding $f \colon \mathcal{X} \to \mathcal{Y}$ and $D$ a convex relaxation. We say that the $D$-analysis is *precise* for $h$ if it yields precise lower and upper bounds, that is for all boxes $\mathcal{B} \subset \mathcal{X}$ we have that $[\underline{h^D(\mathcal{B})}, \overline{h^D(\mathcal{B})}] = [\underline{f(\mathcal{B})}, \overline{f(\mathcal{B})}]$.

In this work, we investigate the expressivity of ReLU-networks, that is, which function class they can encode such that their $D$-analysis is precise. Specifically:

**Definition 5** (Expressivity). Let $D$ be a convex relaxation, $\mathcal{F}$ a set of functions, and $\mathcal{N}$ a set of networks. We say that $\mathcal{N}$ can $D$-*express* $\mathcal{F}$ precisely, if and only if, for all $f \in \mathcal{F}$, there exists a network $h \in \mathcal{N}$, such that $h$ encodes $f$ and its $D$-analysis is precise.

We can often replace (sub-)networks to encode the same function but yield a (strictly) more precise analysis in terms of the obtained input-output polytope:

**Definition 6** (Replacement). Let $h$ and $h'$ be ReLU networks, $\mathcal{B}$ a box, and $D$ some convex relaxation. We say $h'$ can *replace* $h$ with respect to $D$, if $h'^D(\mathcal{B}) \subseteq h^D(\mathcal{B})$ for all $\mathcal{B}$ and write $h \overset{D}{\rightsquigarrow} h'$.

## 3 Related Work

Below, we give a brief overview of the most relevant related work.

**Expressing CPWL Functions** He et al. (2020) show that ReLU networks require at least 2 layers to encode CPWL functions in $\mathbb{R}^d$ (for $d \geq 2$) with $\lceil \log_2(d+1) \rceil$ layers always being sufficient.

**Expressivity with IBP** Baader et al. (2020) show that for any continuous function $f \colon \Gamma \subset \mathbb{R}^n \to \mathbb{R}$ over a compact domain $\Gamma$ and $\epsilon > 0$, there exists a finite ReLU network $h$, such that its IBP-analysis for any input box $\mathcal{B} \subset \Gamma$, denoted by $h^{\mathrm{IBP}}(\mathcal{B})$, is *precise up to an $\epsilon$-error*:

$$[\underline{f(\mathcal{B})} + \epsilon, \overline{f(\mathcal{B})} - \epsilon] \subseteq h^{\mathrm{IBP}}(\mathcal{B}) \subseteq [\underline{f(\mathcal{B})} - \epsilon, \overline{f(\mathcal{B})} + \epsilon].$$

An equivalent result immediately follows for all strictly more precise domains such as DP-0, $\Delta$, and MN. Wang et al. (2022) propose a more efficient construction, generalize this result to squashable activation functions, and provide first results on the hardness of constructing such networks. However, as these results require network widths going to $\infty$ for approximation errors $\epsilon \to 0$ and IBP-based methods fail empirically for realistic networks, the study of exact encodings is crucial.

Investigating what class of functions allows for an *exact* IBP-analysis, Mirman et al. (2022) show that for any function with points of non-invertibility, i.e., $x = 0$ for $f(x) = |x|$, there does not exist a ReLU network IBP-expressing this function.

**Certified Training** Certified training methods typically optimize an upper bound on the worst-case loss over some adversary specification computed via convex relaxations. Surprisingly, using the imprecise IBP-relaxation (Mirman et al., 2018; Gowal et al., 2018) consistently yields better performance than tighter relaxations (Wong et al., 2018; Zhang et al., 2020; Balunović & Vechev, 2020). Jovanović et al. (2022) investigate this paradox and identify two key properties of the worst-case loss approximation, continuity and sensitivity, required for effective optimization, with only IBP possessing both. However, the heavy regularization that makes IBP trained networks amenable to certification also severely reduces their standard accuracy (Mao et al., 2024).

**Neural Network Certification** We distinguish complete certification methods, which, given sufficient time, can decide any property, i.e., always compute precise bounds, and incomplete methods, which sacrifice precision for speed. Salman et al. (2019) unify a range of incomplete certification methods including IBP, DEEPPOLY, and $\Delta$, and show that their precision is limited by that of the $\Delta$-relaxation. They observe that for a wide range of networks and even when using the $\Delta$-relaxation, a substantial certification gap between the upper- and lower-bounds on robust accuracy remains. Semidefinite programming based methods (Dathathri et al., 2020; Raghunathan et al., 2018) increase tightness at the cost of computational efficiency.

Early, complete certification methods directly leveraged off-the-shelf SMT (Katz et al., 2017; Ehlers, 2017) or MILP solvers (Dutta et al., 2018; Tjeng et al., 2019), limiting their applicability to small networks. To improve scalability, Bunel et al. (2020) formulate a branch-and-bound (BaB) framework, that recursively splits the certification problem into easier subproblems until they can be decided by cheap incomplete methods. This concept has been widely adopted and improved using more efficient solvers (Xu et al., 2021; Wang et al., 2021) and tighter constraints (Palma et al., 2021; Ferrari et al., 2022; Zhang et al., 2022).

## 4 CONVEX RELAXATIONS FOR UNIVARIATE FUNCTIONS

In this section, we differentiate all convex relaxations that are commonly used for neural network certification (IBP, DP-0, DP-1, $\Delta$, and MN) in terms of their expressivity, i.e., with respect to the function classes they can analyze precisely when encoded by a ReLU network.

We first show that finite-depth ReLU networks can IBP-express M-CPWL functions precisely (Theorem 9). This construction can be applied directly to the strictly more precise DP-0 and $\Delta$ relaxation and with slight modification also to DP-1. We, then, show that while finite ReLU networks can both DP-0- and $\Delta$-express M-CPWL and C-CPWL functions, the solution space is exponentially larger when using the more precise $\Delta$-analysis. Finally, we show that single-layer ReLU networks can MN-express arbitrary CPWL functions. We defer all proofs and supplementary lemmata to App. B.

### 4.1 BOX

To show that M-CPWL functions can be IBP-expressed, we begin by constructing a step function, illustrated in (Fig. 5), as a two-layer ReLU network that can be IBP-expressed:

**Lemma 7** (Step Function). Let $\beta \in \mathbb{R}_{\geq 0}$ and $f \in$ CPWL$(\mathbb{I}, \mathbb{R})$ s.t. $f(x) = 0$ for $x < x_0$, $f(x) = \beta$ for $x > x_1$ and linear in between. Then, $\phi_{x_0, x_1, \beta}(x) = \beta - \text{ReLU}(\beta - \frac{\beta}{x_1 - x_0} \text{ReLU}(x - x_0))$ encodes $f$.

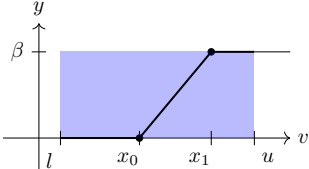

Figure 5: IBP-analysis of the step function $\beta - \text{ReLU}(\beta - \frac{\beta}{x_1 - x_0} \text{ReLU}(x - x_0))$.

**Lemma 8** (Precise Step). The IBP-analysis of $\phi_{x_0,x_1,\beta}$ is precise.

Intuitively, the key to this construction is to leverage that while the IBP-relaxation of the ReLU function does not capture any relational information, it recovers the exact output interval. By using two sequential ReLUs, we allow the inner one to cut away the output-half-space $f(x) < 0$ and the outer one to cut away the half-space $f(x) > \beta$, thus obtaining a precise analysis.

We can now construct arbitrary M-CPWL functions from these step functions, allowing us to show that they too can be IBP-expressed:

**Theorem 9** (Precise Monotone). Finite ReLU networks can IBP-express the set of monotone $\text{CPWL}(\mathbb{I}, \mathbb{R})$ functions precisely.

## 4.2 DEEPPOLY-0

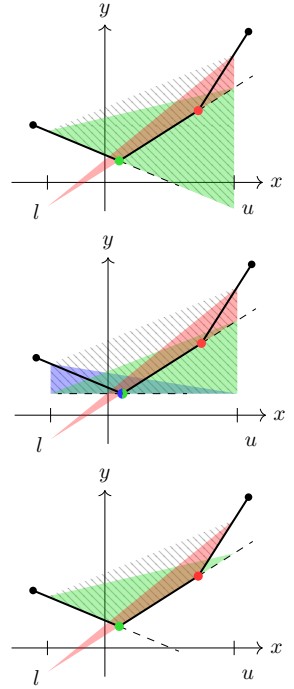

We show constructively that finite ReLU networks can DP-0-express C-CPWL functions, by first encoding any such function as a single-layer ReLU network. We note that the below results equivalently apply to concave functions:

**Lemma 10** (Convex encoding). Let $f \in \text{CPWL}(\mathbb{I}, \mathbb{R})$ be convex. Then $f$ is encoded by

$$h(x) = b + cx + \sum_{i=1}^{n-1} \gamma_i \, \text{ReLU}(\pm_i(x - x_i)), \qquad (1)$$

for any choice $\pm_i \in \{-1, 1\}$, if $b$ and $c$ are set appropriately, where $\alpha_i = \frac{f(x_{i+1}) - f(x_i)}{x_{i+1} - x_i}$ is the slope between points $x_i$ and $x_{i+1}$, and $\gamma_i = \alpha_i - \alpha_{i-1} > 0$ the slope change at $x_{i+1}$.

Intuitively, we encode the C-CPWL function $f$ by starting with a linear function $h_0 = b + cx$, coinciding with one of the linear segments of $f$. We then pick one of the points $x_i$ where $f$ changes slope that are adjacent to this segment and add $\text{ReLU}(\pm_i(x - x_i))$ changing its activation state at this point. Regardless of $\pm_i$, we now scale this ReLU with $\gamma_i = \alpha_i - \alpha_{i-1}$ to introduce the local change of slope, and update the linear term $c \leftarrow c - \gamma_i$ if the newly added ReLU affects the segment that the linear function matched originally. We repeat this process until $h$ encodes $f$.

Figure 6: Illustration of two different ReLU network encodings of the same function unde DP-0- (top and middle) and $\Delta$-analysis (bottom).

We illustrate this in Fig. 6 (top), where we start our construction with the left-most linear segment. We continue by adding a ReLU, first at the green and then the red point, and show the DP-0 relaxation of the added ReLUs as a shaded area of the same color. We illustrate the resulting overall DP-0-relaxation, obtained as their point-wise sum, striped grey. Observe that this always recovers the original linear term as the lower bound (see Fig. 6 top). This leads to an imprecise output range unless its slope $c$ is 0. If $f$ includes such a constant section with zero-slope, we can directly apply the above construction, always changing $\pm_i$ such that the ReLUs open "outward", i.e., in a direction that does not affect the constant segment. If $f$ does not include such a constant section but a unique minimum, as in our example, we place two ReLUs at this point, treating it as a constant section with 0-width and recovering a precise lower bound (see Fig. 6 middle). Thus finite ReLU networks can DP-0-express C-CPWL functions but do not allow $\pm_i$ to be chosen freely. Note that the upper bound is still precise regardless of the choice of $\pm_i$.

**Theorem 11** (DP-0 Convex). For any convex CPWL function $f: \mathbb{I} \to \mathbb{R}$, there exists exactly one network of the form $h(x) = b + \sum_{i \in \mathcal{I}} \gamma_i \, \text{ReLU}(\pm_i(x - x_i))$ encoding $f$, with $|\mathcal{I}| = n - 1$ if $f$ has slope zero on some segment and otherwise $|\mathcal{I}| = n$, such that its DP-0-analysis is precise, where $\gamma_i > 0$ for all $i$.

### 4.3 DEEPPOLY-1

To show that DP-1 has the same expressivity as DP-0, we encode a ReLU function as $h(x) = x + \text{ReLU}(-x)$ which under DP-1-analysis yields the same linear bounds as $h'(x) = \text{ReLU}(x)$ under DP-0-analysis. The reverse also holds. Thus, the expressivity of DP-0 and DP-1 is equivalent.

**Corollary 12** (DP-1 ReLU). The ReLU network $h(x) = x + \text{ReLU}(-x)$ encodes the function $f(x) = \text{ReLU}(x)$ and, the DP-1-analysis of $h(x)$ is identical to the DP-0-analysis of $\text{ReLU}$. Further, the DP-0-analysis of $h(x)$ is identical to the DP-1-analysis of $\text{ReLU}$.

It follows directly that any function that can be DP-0-expressed by a finite ReLU network can be DP-1-expressed by the same ReLU network after substituting every $\text{ReLU}(x)$ with $x + \text{ReLU}(-x)$:

**Corollary 13** (DP-1 Approximation). Finite $\text{ReLU}$ networks can DP-1- and DP-0-express the same function class precisely. In particular, they can DP-1-express the set of convex functions $f \in \text{CPWL}(\mathbb{I}, \mathbb{R})$ and monotone functions $f \in \text{CPWL}(\mathbb{I}, \mathbb{R})$ precisely.

### 4.4 TRIANGLE

To show that finite ReLU networks can $\Delta$-express C-CPWL functions, we reuse the construction from Lemma 10. However, as the $\Delta$-relaxation yields the exact convex hull for ReLU functions, we first show that the convex hull of a sum of convex functions (such as Eq. (1)) is recovered by the pointwise sum of their convex hulls:

**Lemma 14** (Convex Hull Sum). Given two convex functions $f, g \colon \mathbb{R} \to \mathbb{R}$ and the box $[l, u]$. Then, the pointwise sum of the convex hulls $\mathcal{H}_f + \mathcal{H}_g$ is identical to the convex hull of the sum of the two functions $\mathcal{H}_{f+g} = \mathcal{H}_f + \mathcal{H}_g$.

This follows directly from the definition and implies that the $\Delta$-analysis is precise for arbitrary choices of $\pm_i$, illustrated in the bottom of Fig. 6:

**Theorem 15** ($\Delta$ Convex). Let $f \in \text{CPWL}(\mathbb{I}, \mathbb{R})$ be convex. Then, for any network $h$ encoding $f$ as in Lemma 10, we have that its $\Delta$-analysis is precise. In particular, $\pm_i$ can be chosen freely.

### 4.5 MULTI-NEURON-RELAXATIONS

As multi-neuron relaxations yield the exact convex hull of the considered group of neurons (all within the same layer), it is sufficient to show that we can express arbitrary CPWL functions with a single-layer network to see that they MN-express CPWL functions. To this end, we use a similar construction as in Lemma 10, where the lack of convexity removes the positivity constraint on $\gamma_i$.

**Theorem 16** (Multi-Neuron Precision). For every $f \in \text{CPWL}(\mathbb{I}, \mathbb{R})$, there exists a single layer ReLU network $h$ encoding $f$, such that its MN-analysis (considering all ReLUs jointly) is precise.

## 5 CONVEX RELAXATIONS FOR MULTIVARIATE FUNCTIONS

After having shown in the previous section that, for univariate functions, the expressivity of ReLU networks under convex relaxations is not fundamentally limited, we now turn our attention to multivariate functions. There, we prove that no finite ReLU network can $\Delta$-express the maximum function $\max \colon \mathbb{R}^2 \to \mathbb{R}$ precisely (Theorem 20). This directly implies that no single-neuron relaxation can express the class of multivariate, monotone, and convex CPWL functions precisely. Note, this negative result generalizes to all relaxations less precise than $\Delta$, including IBP and DEEPPOLY.

Intuitively, we will argue along the following lines. We first observe that for any finite ReLU-Network $h$ that encodes the maximum function, we can find a point $(x, y = x) \in \mathbb{R}^2$ with neighborhood $\mathcal{U}$, such that on $\mathcal{U}$, all ReLUs in $h$ either switch their activation state for $x = y$ or not at all (see Fig. 7). Then,

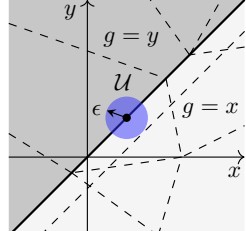

Figure 7: Illustration of the pre-image of activation pattern changes with (—) and without (- -) functional change of a ReLU network $h$ encoding the $g = \max(x, y)$ function, as well as the $\epsilon$-neighborhood $\mathcal{U}$ (●), in which the only activation change occurs at $x = y$.

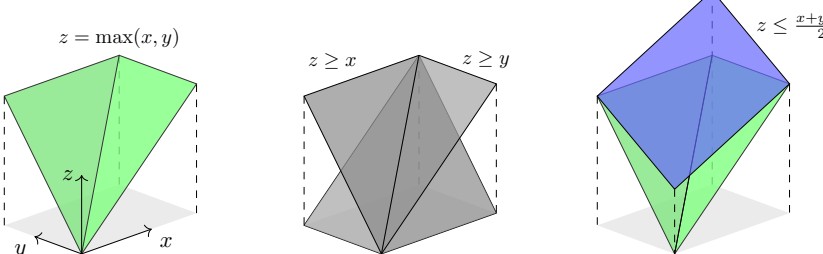

Figure 8: Illustration of $z = \max(x, y)$ (left) with its $\Delta$ lower (middle) and upper (right) bounds as obtained in Theorem 19.

we show that for such a neighborhood, we can $\Delta$-replace the finite ReLU network $h$ with a single layer consisting of just 2 neurons and a linear term (Theorems 17 and 18). Finally, we show that no such single-layer network can $\Delta$-express $\max$ precisely (Theorem 19), before putting everything together in Theorem 20. All proofs and support lemmata are again deferred to App. A.

Let us begin by showing that we can express *any* finite ReLU network using the functional form of Eq. (2). That is, every $i$-layer network $\boldsymbol{h}^i$ can be written as the sum of an $(i-1)$-layer network $\boldsymbol{h}_L^{i-1}$ and a linear function of a ReLU applied to another $(i-1)$-layer network $\boldsymbol{W}_i \operatorname{ReLU}(\boldsymbol{h}_R^{i-1})$. Further, if, for a given input region $\mathcal{U}$, all ReLUs in the original network switch activation state on the hyperplane $\boldsymbol{w}^\top \boldsymbol{x} = 0$ or not at all, then, we can ensure that *every* ReLU in both $(i-1)$-layer networks change activation state exactly for $z \coloneqq \boldsymbol{w}^\top \boldsymbol{x} = 0$.

**Theorem 17** (Network Form Coverage). Given a neighborhood $\mathcal{U}$ and a finite $k$-layer ReLU network $h$ such that on $\mathcal{U}$ and under $\Delta$-analysis all its ReLUs are either stably active ($\operatorname{ReLU}(v) = v$), stably inactive ($\operatorname{ReLU}(v) = 0$), or switch activation state for $z \coloneqq \boldsymbol{w}^\top \boldsymbol{x} = 0$ with $\boldsymbol{w} \in \mathbb{R}^d$, then $h$ can be represented using the functional form

$$\boldsymbol{h}_{\{R,L\}}^i = \boldsymbol{h}_L^{i-1} + \boldsymbol{W}_i \operatorname{ReLU}(\boldsymbol{h}_R^{i-1}), \quad \boldsymbol{h}_{\{R,L\}}^0 = \boldsymbol{b} + \boldsymbol{W}_0 \boldsymbol{x}, \tag{2}$$

for $i = k$ and such that all ReLUs switch their activation state at $\{\boldsymbol{x} \in \mathcal{X} \mid \boldsymbol{w}^\top \boldsymbol{x} = 0\}$. Here, $L$ and $R$ are labels, used to distinguish the possibly different (i-1)-layer networks $\boldsymbol{h}^{i-1}$ from each other.

We can now leverage the fact that all ReLUs change activation state at the same point to simplify the sum of ReLUs to a linear term plus a single ReLU: $\sum_i a_i \operatorname{ReLU}(w_i z) \overset{\Delta}{\rightsquigarrow} \gamma z + \alpha \operatorname{ReLU}(z)$ for some $\gamma, \alpha \in \mathbb{R}$ (Lemma 22). This allows us to further simplify a ReLU applied to such a sum of ReLUs: $\operatorname{ReLU}(\gamma + \alpha \operatorname{ReLU}(z)) \overset{\Delta}{\rightsquigarrow} \gamma' z + \alpha' \operatorname{ReLU}(z)$ (Lemma 23). These two replacements allow us to recursively reduce the depth of networks in the form of Eq. (2) until just a single layer is left:

**Theorem 18** (Network Simplification). Let $h^k$ be a network as in Theorem 17 such that all ReLUs change activation state at $z \coloneqq \boldsymbol{w}^\top \boldsymbol{x} = 0$ with $\boldsymbol{w} \in \mathbb{R}^d$. We have

$$h^k = h_L^{k-1} + \boldsymbol{W} \operatorname{ReLU}(\boldsymbol{h}_R^{k-1}) \quad \overset{\Delta}{\rightsquigarrow} \quad h(\boldsymbol{x}) = b + \boldsymbol{W} \boldsymbol{x} + \alpha \operatorname{ReLU}(z),$$

where $h^0(\boldsymbol{x}) = b_0 + \boldsymbol{W}_0 \boldsymbol{x}$ and all ReLU change state exactly at $\{\boldsymbol{x} \in \mathcal{X} \mid \boldsymbol{w}^\top \boldsymbol{x} = 0\}$.

Note that, $h^k$, $h_L^{k-1}$ and $h$ map to $\mathbb{R}$, while $\boldsymbol{h}_R^{k-1}$ maps to the space of some hidden layer $\mathbb{R}^n$. Next, we show directly via contradiction that single-layer ReLU networks of this form cannot $\Delta$-express the maximum function, illustrating the resulting (imprecise) bounds in Fig. 8:

**Theorem 19** (Triangle max). ReLU-networks of the form $h(x, y) = b + w_x x + w_y y + \alpha \operatorname{ReLU}(x - y)$ can not $\Delta$-express the function $\max \colon \mathbb{R}^2 \to \mathbb{R}$.

*Proof.* We consider the input region $\mathcal{B} = [0, 1]^2$ and constrain our parameters by considering the following: For $x = y = 0$, we have $f(0, 0) = 0 = b = h(0, 0)$. For $x < y$, we have $f(x, y) = y = w_x x + w_y y = h(x, y)$ and thus $w_x = 0$, $w_y = 1$. Finally for $x > y$, we have $f(x, y) = x = y + \alpha(x - y) = h(x, y)$ and thus $\alpha = 1$. Hence we have $h(x, y) = y + \operatorname{ReLU}(x - y)$.

$$\left.\begin{array}{c} 0 \\ x - y \end{array}\right\} \le \operatorname{ReLU}(x - y) \le \tfrac{1}{2}(x - y + 1) \quad \implies \quad \left.\begin{array}{c} y \\ x \end{array}\right\} \le h(x, y) \le \tfrac{1}{2}(x + y + 1).$$

The maximum of the upper bound is attained at $x = y = 1$, where we get $\overline{h^\Delta(\mathcal{B})} = \tfrac{3}{2}$ which is larger than $\overline{\max(\mathcal{B})} = 1$ (see Fig. 8). $\qquad \square$

To show that no ReLU network can $\Delta$-express $\max$ it remains to argue how we can find a $\mathcal{U}$ such that all ReLUs switch their activation state at $\{\boldsymbol{x} \in \mathcal{X} \mid \boldsymbol{w}^\top \boldsymbol{x} = 0\}$ or not at all:

**Theorem 20** ($\Delta$ Impossibility $\max$). Finite ReLU networks can not $\Delta$-express the function $\max$.

*Proof.* We will prove this theorem via contradiction in four steps. Assume there exists a finite ReLU network $h$ that $\Delta$-expresses $\max$ precisely.

**First – Locality** We argue this point in three steps:

1. There exists a point $(x, y = x)$ with an $\epsilon$-neighborhood $\mathcal{U}'$ such that one of the following holds for any $\mathrm{ReLU}(v)$ with input $v = h_v(x, y)$ of the network $h$:

   - the ReLU is always active, i.e., $\forall (x, y) \in U, \mathrm{ReLU}(v) = v$,
   - the ReLU is never active, i.e., $\forall (x, y) \in U, \mathrm{ReLU}(v) = 0$, or
   - the ReLU changes activation state for $x = y$, i.e., $\exists v' \in \mathbb{R}, s.t., \mathrm{ReLU}(v) = \mathrm{ReLU}(v'(x - y))$.

   This follows directly from the fact that finite ReLU networks divide their input space into finitely many linear regions (see the illustration in Fig. 7).

2. Further, there exists an $\epsilon$-neighborhood $\mathcal{U}$ of $(x, y)$ such that the above holds under IBP-analysis, as it depends continuously on the input bounds and becomes exact for any network when the input bounds describe a point.

3. Via rescaling and translation, we can assume that the point $(x, y)$ is at $\boldsymbol{0}$ and that the neighborhood $\mathcal{U}$ covers $[-1, 1]^2$.

**Second – Network Form** On the neighborhood $\mathcal{U}$, any finite ReLU-network $h$ can, w.r.t. $\Delta$, be represented by $h^k = h^{k-1} + \boldsymbol{W} \mathrm{ReLU}(\boldsymbol{h}^{k-1}), h^0(v) = \boldsymbol{b} + \boldsymbol{W}\boldsymbol{v}$ with biases $\boldsymbol{b} \in \mathbb{R}^{d_k}$, weight matrices $\boldsymbol{W} \in \mathbb{R}^{d_k \times d_{k-1}}$, where all ReLUs change activation state exactly for $x = y$ (Theorem 17).

**Third – Network Replacements** We can replace $h^k$ w.r.t. $\Delta$ with the single layer network $h'(x, y) = b + \boldsymbol{W}(x, y)^\top + \alpha R(x - y)$ (Theorem 18).

**Fourth – Conclusion** There exists no network of this form encoding the $\max$-function such that its $\Delta$-analysis is precise on the interval $[0, 1]^2$ (Theorem 19).

This concludes the proof. $\square$

As $\max$ belongs to the class of multivariate, convex, monotone, CPWL functions, it follows directly from Theorem 20 that no finite ReLU network can $\Delta$-express this class precisely:

**Corollary 21** ($\Delta$ Impossibility). Finite ReLU networks can not $\Delta$-express the set of convex, monotone, CPWL functions mapping from some box $\mathbb{I} \subset \mathbb{R}^2$ to $\mathbb{R}$.

## 6 CONCLUSION

We conduct the first in-depth study on the expressivity of ReLU networks under all commonly used convex relaxations and find that: (i) more precise relaxations ($\Delta$, DP-0 or DP-1) allow a larger class of *univariate* functions (C-CPWL and M-CPWL) to be expressed precisely than the simple IBP-relaxation (M-CPWL), (ii) for the same function class (C-CPWL), a more precise relaxation ($\Delta$ vs DP-0 or DP-1), can allow an exponentially larger solution space of ReLU networks, (iii) MN-relaxations allow single-layer networks to express all univariate CPWL functions, (iv) even the most precise single-neuron relaxation ($\Delta$) is too imprecise to express *multivariate*, convex, monotone CPWL functions precisely with finite ReLU networks, despite their exact analysis being trivial.

While more precise domains improve expressivity for univariate functions, all single-neuron convex-relaxations are fundamentally limited in the multivariate setting. Surprisingly, even simple functions that can be encoded with a single neuron $h = y + \mathrm{ReLU}(x - y) = \max(x, y)$, can not be $\Delta$-expressed precisely using any finite ReLU network. This highlights not only the importance of recent, more precise multi-neuron- and BaB-based neural network certification methods but also suggests more precise methods might be needed for training.

ACKNOWLEDGEMENTS

We would like to thank our anonymous reviewers for their constructive comments and insightful questions.

This work has been done as part of the EU grant ELSA (European Lighthouse on Secure and Safe AI, grant agreement no. 101070617). Views and opinions expressed are however those of the authors only and do not necessarily reflect those of the European Union or European Commission. Neither the European Union nor the European Commission can be held responsible for them.

The work has received funding from the Swiss State Secretariat for Education, Research and Innovation (SERI).

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

## A    DEFERRED PROOFS ON MULTIVARIATE FUNCTIONS

**Theorem 17** (Network Form Coverage). Given a neighborhood $\mathcal{U}$ and a finite $k$-layer ReLU network $h$ such that on $\mathcal{U}$ and under $\Delta$-analysis all its ReLUs are either stably active ($\mathrm{ReLU}(v) = v$), stably inactive ($\mathrm{ReLU}(v) = 0$), or switch activation state for $z := \boldsymbol{w}^\top \boldsymbol{x} = 0$ with $\boldsymbol{w} \in \mathbb{R}^d$, then $h$ can be represented using the functional form

$$\boldsymbol{h}^i_{\{R,L\}} = \boldsymbol{h}^{i-1}_L + \boldsymbol{W}_i \, \mathrm{ReLU}(\boldsymbol{h}^{i-1}_R), \quad \boldsymbol{h}^0_{\{R,L\}} = \boldsymbol{b} + \boldsymbol{W}_0 \boldsymbol{x}, \tag{2}$$

for $i = k$ and such that all ReLUs switch their activation state at $\{\boldsymbol{x} \in \mathcal{X} \mid \boldsymbol{w}^\top \boldsymbol{x} = 0\}$. Here, $L$ and $R$ are labels, used to distinguish the possibly different (i-1)-layer networks $\boldsymbol{h}^{i-1}$ from each other.

*Proof.* Given $\boldsymbol{W}_i \, \mathrm{ReLU}(\boldsymbol{h}^{i-1}_R)$, we partition the columns of the weight matrix into $\boldsymbol{W}_i = (\boldsymbol{W}^+_i | \boldsymbol{W}^-_i | \boldsymbol{W}^\pm_i)$, depending on whether the associated ReLU is stably active, stably inactive, or unstable, respectively. We thus obtain

$$(\boldsymbol{W}^+_i | \boldsymbol{W}^-_i | \boldsymbol{W}^\pm_i) \, \mathrm{ReLU}(\boldsymbol{h}^{k-1}_R) = \boldsymbol{W}^+_i \boldsymbol{h}^{k-1}_R + \boldsymbol{W}^\pm_i \, \mathrm{ReLU}(\boldsymbol{h}^{k-1}_R).$$

We update $\boldsymbol{h}^{i-1}_{L,new} = \boldsymbol{h}^{i-1}_L + \boldsymbol{W}^+_i \boldsymbol{h}^{i-1}_R$, by showing that $\boldsymbol{W}^+_i \boldsymbol{h}^{i-1}_R$ is still an $(i-1)$-layer network as follows. We recursively update weight matrices $\boldsymbol{W}_{k,new} = \boldsymbol{W}^+_i \boldsymbol{W}_k$ to obtain $\boldsymbol{W}^+_i \boldsymbol{h}^k = \boldsymbol{W}^+_i \boldsymbol{h}^{k-1} + \boldsymbol{W}_{k,new} \, \mathrm{ReLU}(\boldsymbol{h}^{k-1}_R)$ until we have reached $k = 1$, where we have $\boldsymbol{W}^+_i \boldsymbol{h}^1 = \boldsymbol{W}^+_i \boldsymbol{b}_L + \boldsymbol{W}^+_i \boldsymbol{W}_{0,L} \boldsymbol{v} + \boldsymbol{W}_{1,new} \, \mathrm{ReLU}(\boldsymbol{b}_R + \boldsymbol{W}_{0,R} \boldsymbol{x})$. $\qquad\square$

**Lemma 22** (Simplification of ReLU Sums w.r.t. $\Delta$). Let $\boldsymbol{A} \in \mathbb{R}^{n \times 1}$ and $\boldsymbol{w} \in \mathbb{R}^n$. Then, we have

$$h(z) = \boldsymbol{A}^\top \mathrm{ReLU}(\boldsymbol{w}z) \quad \overset{\Delta}{\rightsquigarrow} \quad h'(z) = \gamma z + \alpha \, \mathrm{ReLU}(z),$$

where $\gamma = \sum_{i,w_i < 0} A_i w_i$ and $\alpha = \sum_{i,w_i > 0} A_i w_i - \gamma$.

*Proof.* Both $h$ and $h'$ are CPWL functions with slope change only at $z = 0$. Thus they are fully defined by their value at the points $z_i \in \{-1, 0, 1\}$. Hence, we can show that $h$ and $h'$ encode the same function by showing their equivalence on these points: $h(0) = 0 = h'(0)$, $h(-1) = \sum_{i,w_i < 0} -A_i w_i = -\gamma = h'(-1)$, and $h(1) = \sum_{i,w_i > 0} A_i w_i = \alpha + \gamma$. As $\gamma z$ and $\alpha \, \mathrm{ReLU}(z)$ are convex/concave, and their $\Delta$-analysis yields their convex hulls, the pointwise sum of their convex hulls, i.e. the $\Delta$-analysis of $h'$, recovers the convex hull of $h'$ by Lemma 14 and is thus at least as precise as any convex-relaxation of $h$. $\qquad\square$

**Lemma 23** (Simplification of Composed ReLUs w.r.t. $\Delta$). We have

$$h(z) = \mathrm{ReLU}(\gamma z + \alpha \, \mathrm{ReLU}(z)) \quad \overset{\Delta}{\rightsquigarrow} \quad h'(z) = \gamma' z + \alpha' \, \mathrm{ReLU}(z),$$

where $\gamma' = -\mathrm{ReLU}(-\gamma)$ and $\alpha' = \mathrm{ReLU}(\alpha + \gamma) - \gamma'$.

*Proof.* We observe that $h(z)$ is convex and piecewise-linear for any $z \in [l, u] \subset \mathbb{R}$ with $l < 0 < u$ and a slope change only at $z = 0$. Its convex hull is thus spanned by $h(l) = \mathrm{ReLU}(\gamma l) = h'(l)$, $h(0) = h'(0) = 0$, and $h(u) = \mathrm{ReLU}((\gamma + \alpha)u) = h'(u)$. We further observe that the $\Delta$-relaxation of $\mathrm{ReLU}(z)$ and $z$ is their convex hull. Finally, the convex hull of the positive sum of convex functions is equal to the pointwise sum of their individual convex hulls (Lemma 14). Thus, the triangle-relaxation of $h'(z)$ recovers the convex hull of $h(z)$ and thus the tightest possible convex relaxation. $\qquad\square$

For convex functions $f, g \colon \mathbb{R} \to \mathbb{R}$, we define the convex hull $\mathcal{H}_f([l,u]) = \{(x,y) \mid x \in [l,u], f(x) \le y \le f(l) + \frac{f(u)-f(l)}{u-l}(x-l)\}$ over $[l,u] \subset \mathbb{R}$. Further, we define the convex hull sum of $f$ and $g$ on $[l,u]$ to be $\mathcal{H}_f + \mathcal{H}_g := \{(x, y' + y'') \mid (x,y') \in \mathcal{H}_f, (x,y'') \in \mathcal{H}_g\}$.

**Lemma 14** (Convex Hull Sum). Given two convex functions $f, g \colon \mathbb{R} \to \mathbb{R}$ and the box $[l,u]$. Then, the pointwise sum of the convex hulls $\mathcal{H}_f + \mathcal{H}_g$ is identical to the convex hull of the sum of the two functions $\mathcal{H}_{f+g} = \mathcal{H}_f + \mathcal{H}_g$.

*Proof.* We first show that every point in $\mathcal{H}_{f+g}$ can be obtained from $\mathcal{H}_f + \mathcal{H}_g$. Let $(x,y) \in \mathcal{H}_{f+g}([l,u])$. Then we have

$$(f+g)(x) \le y \le (f+g)(l) + \tfrac{(f+g)(u)-(f+g)(l)}{u-l}(x-l),$$

$$f(x) + g(x) \le y \le f(l) + \tfrac{f(u)-f(l)}{u-l}(x-l) + g(l) + \tfrac{f(u)-f(l)}{u-l}(x-l).$$

Then we can find a partition of $y = y' + y''$. We know for sure that there exists $t \in [0, 1]$ s.t.

$$y = (1 - t)(f + g)(x) + t((f + g)(l) + \frac{(f+g)(u) - (f+g)(l)}{u-l}(x - l)).$$

Hence if we pick for example

$$y' = (1 - t)f(x) + t(f(l) + \frac{f(u)-f(l)}{u-l}(x - l)) \in \mathcal{H}_f$$

$$y'' = (1 - t)g(x) + t(g(l) + \frac{g(u)-g(l)}{u-l}(x - l)) \in \mathcal{H}_g,$$

we get immediately that $(x, y) \in \mathcal{H}_f + \mathcal{H}_g$. The other direction is immediate. $\square$

Using Lemma 23, we can show that these networks mapping $\mathbb{R}^2$ to $\mathbb{R}$ can be simplified further:

**Theorem 18** (Network Simplification). Let $h^k$ be a network as in Theorem 17 such that all ReLUs change activation state at $z := \boldsymbol{w}^\top \boldsymbol{x} = 0$ with $\boldsymbol{w} \in \mathbb{R}^d$. We have

$$h^k = h_L^{k-1} + \boldsymbol{W} \operatorname{ReLU}(\boldsymbol{h}_R^{k-1}) \quad \overset{\Delta}{\leadsto} \quad h(\boldsymbol{x}) = b + \boldsymbol{W}\boldsymbol{x} + \alpha \operatorname{ReLU}(z),$$

where $h^0(\boldsymbol{x}) = b_0 + \boldsymbol{W}_0\boldsymbol{x}$ and all ReLU change state exactly at $\{\boldsymbol{x} \in \mathcal{X} \mid \boldsymbol{w}^\top \boldsymbol{x} = 0\}$.

Note that, $h^k$, $h_L^{k-1}$ and $h$ map to $\mathbb{R}$, while $\boldsymbol{h}_R^{k-1}$ maps to some $\mathbb{R}^n$.

*Proof.* We show a more general result on $\boldsymbol{h}^k$ with possibly many output dimensions by induction:

*Induction Hypothesis:* $\boldsymbol{h}^i \overset{\Delta}{\leadsto} \boldsymbol{b}_i + \boldsymbol{W}_i\boldsymbol{x} + \boldsymbol{\alpha}_i \operatorname{ReLU}(z)$.

*Base Case:* $\boldsymbol{h}^0(\boldsymbol{x}) = \boldsymbol{b}_0 + \boldsymbol{W}_0\boldsymbol{x}$ satisfies the form $\boldsymbol{h}^0(\boldsymbol{x}) = \boldsymbol{b}_0 + \boldsymbol{W}_0\boldsymbol{x} + \boldsymbol{\alpha}_0 \operatorname{ReLU}(z)$ for $\boldsymbol{\alpha}_0 = \boldsymbol{0}$, thus we can replace $\boldsymbol{h}^0(\boldsymbol{x})$ by itself.

*Induction Step:* Using the induction hypothesis, we have $\boldsymbol{W}_i \operatorname{ReLU}(\boldsymbol{h}_R^{i-1}) = \boldsymbol{W}_i \operatorname{ReLU}(\boldsymbol{b}_{i-1} + \boldsymbol{W}_{i-1}\boldsymbol{x} + \boldsymbol{\alpha}_{i-1} \operatorname{ReLU}(z))$, which by Theorem 17 only changes its activation state at $z = 0$. Since $\operatorname{ReLU}(0) = 0$, we must have $\boldsymbol{b}_{i-1} + \boldsymbol{W}_{i-1}\boldsymbol{x} = \boldsymbol{w}z$ for some $\boldsymbol{w}$ (recall that $z$ is the projection of $\boldsymbol{x}$ on a hyperplane in the input space). Further, applying Lemma 23, we obtain

$$\boldsymbol{W}_i \operatorname{ReLU}(\boldsymbol{h}_R^{i-1}) = \boldsymbol{W}_i \operatorname{ReLU}(\boldsymbol{w}z + \boldsymbol{\alpha}_{i-1} \operatorname{ReLU}(z))$$

$$\overset{\Delta}{\leadsto} \boldsymbol{\gamma}_i' z + \boldsymbol{\alpha}_i' \operatorname{ReLU}(z) = \boldsymbol{b}_i' + \boldsymbol{W}_i'\boldsymbol{x} + \boldsymbol{\alpha}_i' \operatorname{ReLU}(z).$$

Using the induction hypothesis, we can thus rewrite:

$$\boldsymbol{h}^i = \boldsymbol{h}^{i-1} + \boldsymbol{W}_i \operatorname{ReLU}(\boldsymbol{h}_R^{i-1}) \overset{\Delta}{\leadsto} \boldsymbol{b} + \boldsymbol{W}\boldsymbol{x} + \boldsymbol{\alpha}_i \operatorname{ReLU}(z).$$

$\square$

**Theorem 19** (Triangle max). ReLU-networks of the form $h(x, y) = b + w_x x + w_y y + \alpha \operatorname{ReLU}(x - y)$ can not $\Delta$-express the function $\max \colon \mathbb{R}^2 \to \mathbb{R}$.

*Proof.* We first constrain our parameters by considering the following:

- For $x = y = 0$, we have $f(0, 0) = 0$, leading to $b = 0 = h(0, 0)$.

- For $x < y$, we have $f(x, y) = y = w_x x + w_y y = h(x, y)$ and thus $w_x = 0$, $w_y = 1$.

- For $x > y$, we have $f(x, y) = x = y + \alpha(x - y) = h(x, y)$ and thus $\alpha = 1$.

Hence we have $h(x, y) = y + \operatorname{ReLU}(x - y)$.

$$\left.\begin{array}{c} 0 \\ x - y \end{array}\right\} \leq \operatorname{ReLU}(x - y) \leq \tfrac{1}{2}(x - y + 1)$$

Adding $y$ results in the following:

$$\left.\begin{array}{c} y \\ x \end{array}\right\} \leq h(x, y) \leq \tfrac{1}{2}(x + y + 1).$$

The maximum of the upper bound is attained at $x = y = 1$, where we get $\frac{3}{2}$ which is larger than $\max(x, y) = 1$ for $x, y \in [0, 1]$.

$\square$

**Theorem 24** (Triangle $\max \ell_p$)**.** ReLU-networks of the form $h(x,y) = b + w_x x + w_y y + \alpha \operatorname{ReLU}(x-y)$ can not $\Delta$-express the function $\max \colon \mathbb{R}^2 \to \mathbb{R}$ for any $\ell_p$-norm bounded perturbation with $p \geq 1$, i.e., input regions $\mathcal{B}_p^\epsilon(x) := \{ \boldsymbol{x}' \in \mathcal{X} \mid \|\boldsymbol{x} - \boldsymbol{x}'\|_p \leq \epsilon \}$.

*Proof.* We first consider the case of $p > 1$ and $f = \max(x,y)$: We again constrain our parameters as in the proof of Theorem 19 to obtain $h(x,y) = y + \operatorname{ReLU}(x-y)$. We consider the input region $\mathcal{B}_p^{\epsilon=0.5}\left( \begin{bmatrix} x_0 \\ y_0 \end{bmatrix} = \begin{bmatrix} 0.5 \\ 0.5 \end{bmatrix} \right)$. We can now use Hölder's inequality to compute bounds on the ReLU input:

$$-\frac{1}{2} \underbrace{\left\| \begin{bmatrix} 1 \\ -1 \end{bmatrix} \right\|_q}_{c_q :=} = x_0 - y_0 - \left\| \begin{bmatrix} 1 \\ -1 \end{bmatrix} \right\|_q \epsilon \leq x - y \leq x_0 - y_0 + \left\| \begin{bmatrix} 1 \\ -1 \end{bmatrix} \right\|_q \epsilon = \frac{1}{2} \underbrace{\left\| \begin{bmatrix} 1 \\ -1 \end{bmatrix} \right\|_q}_{c_q :=},$$

where $\frac{1}{p} + \frac{1}{q} = 1$. And thus obtain the following bounds on the ReLU output:

$$\left. \begin{array}{c} 0 \\ x - y \end{array} \right\} \leq \operatorname{ReLU}(x-y) \leq \tfrac{1}{2}(x - y + \tfrac{c_q}{2}).$$

Adding $y$ results in the following:

$$\left. \begin{array}{c} y \\ x \end{array} \right\} \leq h(x,y) \leq \tfrac{1}{2}(x + y + \tfrac{c_q}{2}).$$

We can again use Hölder's inequality to bound the upper bound:

$$\tfrac{1}{2}(x + y + \tfrac{c_q}{2}) \leq \tfrac{1}{2}(x_0 + y_0 + \tfrac{c_q}{2}) + \underbrace{\left\| \begin{bmatrix} 1 \\ 1 \end{bmatrix} \right\|_q}_{=c_q} \tfrac{\epsilon}{2} = \tfrac{1}{2} + \tfrac{1}{2}c_q = \tfrac{1}{2}(1 + 2^{\frac{1}{q}}) > 1,$$

Where the last inequality holds due to $p > 1 \implies q < \infty$. This upper bound is strictly greater than $\max_{\mathcal{B}_p^{0.5}} \max(x,y) = 1$.

We now consider the case of $p = 1$ and the rotated max function $f = \max(\frac{x-y}{\sqrt{2}}, \frac{x+y}{\sqrt{2}})$. We choose the input region $\mathcal{B}_{p=1}^{\epsilon=\frac{1}{\sqrt{2}}}\left( \begin{bmatrix} \frac{1}{\sqrt{2}} \\ 0 \end{bmatrix} \right)$ and observe that we recover the setting as discussed in the proof of Theorem 19 for $p = \infty$ rotated by $45°$ around the origin and thus obtain the same imprecise bounds. $\qquad\square$

**Theorem 20** ($\Delta$ Impossibility $\max$)**.** Finite ReLU networks can not $\Delta$-express the function $\max$.

*Proof.* We will prove this theorem in four steps.

**First – Locality**

1. There exists a point $(x, y = x)$ with an $\epsilon$-neighborhood $\mathcal{U}'$ such that one of the following holds for any $\operatorname{ReLU}(v)$ with input $v = h_v(x,y)$ of the network $h$:

   - the ReLU is always active, i.e., $\forall (x,y) \in U, \operatorname{ReLU}(v) = v$,
   - the ReLU is never active, i.e., $\forall (x,y) \in U, \operatorname{ReLU}(v) = 0$, or
   - the ReLU changes activation state for $x = y$, i.e., $\exists v' \in \mathbb{R}, s.t., \operatorname{ReLU}(v) = \operatorname{ReLU}(v'(x-y))$.

2. Further, there exists a neighborhood $\mathcal{U}$ of $(x,y)$ such that the above holds under $\Delta$-analysis, as it depends continuously on the input bounds and becomes exact for any network when the input bounds describe a point.

3. Via rescaling and translation, we can assume that the point $(x,y)$ is at $\boldsymbol{0}$ and that the neighborhood $\mathcal{U}$ covers $[-1,1]^2$.

**Second – Network Form** On the neighborhood $\mathcal{U}$, any finite ReLU-network $h$ can, w.r.t. $\Delta$, be replaced by $\boldsymbol{h}^k = \boldsymbol{h}^{k-1} + \boldsymbol{W} \operatorname{ReLU}(\boldsymbol{h}^{k-1})$ with biases $\boldsymbol{b} \in \mathbb{R}^{d_k}$, weight matrices $\boldsymbol{W} \in \mathbb{R}^{d_k \times d_{k-1}}$, and $h^0(v) = \boldsymbol{b} + \boldsymbol{W} \boldsymbol{v}$, where all ReLUs change activation state exactly for $x = y$ (Theorem 17).

**Third – Network Simplifications** We can replace $h^k$ w.r.t. triangle with $b + W_k(x, y)^\top + \alpha_k R(x - y)$ (Theorem 18).

**Fourth – Conclusion** Every finite ReLU network can be replaced w.r.t. $\Delta$ with a single layer network of the form $h^1(x, y) = b + \boldsymbol{W}(x, y)^\top + \alpha R(x - y)$. However, there exists no such network encoding the max-function such that its $\Delta$-analysis is precise on the interval $[0, 1]^2$ (Theorem 19).
□

**Corollary 21** ($\Delta$ Impossibility)**.** Finite ReLU networks can not $\Delta$-express the set of convex, monotone, CPWL functions mapping from some box $\mathbb{I} \subset \mathbb{R}^2$ to $\mathbb{R}$.

# B  DEFERRED PROOFS ON UNIVARIATE FUNCTIONS

## B.1  BOX

**Lemma 7** (Step Function)**.** Let $\beta \in \mathbb{R}_{\geq 0}$ and $f \in \text{CPWL}(\mathbb{I}, \mathbb{R})$ s.t. $f(x) = 0$ for $x < x_0$, $f(x) = \beta$ for $x > x_1$ and linear in between. Then, $\phi_{x_0, x_1, \beta}(x) = \beta - \text{ReLU}(\beta - \frac{\beta}{x_1 - x_0} \text{ReLU}(x - x_0))$ encodes $f$.

*Proof.* We prove the theorem by considering the three cases separately:

1. For $x \leq x_0$ we have

$$\begin{aligned}
\phi_{x_0, x_1, \beta}(x) &= -\text{ReLU}(-\tfrac{\beta}{x_1 - x_0} \text{ReLU}(x - x_0) + \beta) + \beta \\
&= -\text{ReLU}(-\tfrac{\beta}{x_1 - x_0} \cdot 0 + \beta) + \beta \\
&= -\text{ReLU}(\beta) + \beta \\
&= -\beta + \beta \\
&= 0.
\end{aligned}$$

2. For $x_0 \leq x \leq x_1$ we have

$$\begin{aligned}
\phi_{x_0, x_1, \beta}(x) &= -\text{ReLU}(-\tfrac{\beta}{x_1 - x_0} \text{ReLU}(x - x_0) + \beta) + \beta \\
&= -\text{ReLU}(-\tfrac{\beta}{x_1 - x_0}(x - x_0) + \beta) + \beta \\
&= \tfrac{\beta}{x_1 - x_0}(x - x_0) - \beta + \beta \\
&= \tfrac{\beta}{x_1 - x_0}(x - x_0).
\end{aligned}$$

3. For $x \geq x_1$ we have

$$\begin{aligned}
\phi_{x_0, x_1, \beta}(x) &= -\text{ReLU}(-\tfrac{\beta}{x_1 - x_0} \text{ReLU}(x - x_0) + \beta) + \beta \\
&= -\text{ReLU}(-\tfrac{\beta}{x_1 - x_0}(x - x_0) + \beta) + \beta \\
&= -0 + \beta \\
&= \beta.
\end{aligned}$$

□

**Lemma 8** (Precise Step)**.** The IBP-analysis of $\phi_{x_0, x_1, \beta}$ is precise.

*Proof.* Consider the box $[l, u] \subseteq \mathbb{R}$.

$$
\begin{aligned}
\phi_{x_0,x_1,\beta}([l,u]) &= -\operatorname{ReLU}(-\tfrac{\beta}{x_1-x_0}\operatorname{ReLU}([l,u]-x_0)+\beta)+\beta \\
&= -\operatorname{ReLU}(-\tfrac{\beta}{x_1-x_0}\operatorname{ReLU}([l-x_0,u-x_0])+\beta)+\beta \\
&= -\operatorname{ReLU}(-\tfrac{\beta}{x_1-x_0}[\operatorname{ReLU}(l-x_0),\operatorname{ReLU}(u-x_0)]+\beta)+\beta \\
&= -\operatorname{ReLU}([\tfrac{\beta}{x_1-x_0}\operatorname{ReLU}(u-x_0),\tfrac{\beta}{x_1-x_0}\operatorname{ReLU}(l-x_0)]+\beta)+\beta \\
&= -\operatorname{ReLU}([\tfrac{\beta}{x_1-x_0}\operatorname{ReLU}(u-x_0)+\beta,\tfrac{\beta}{x_1-x_0}\operatorname{ReLU}(l-x_0)+\beta])+\beta \\
&= -\operatorname{ReLU}([\tfrac{\beta}{x_1-x_0}\operatorname{ReLU}(u-x_0)+\beta,\tfrac{\beta}{x_1-x_0}\operatorname{ReLU}(l-x_0)+\beta])+\beta \\
&= -[\operatorname{ReLU}(\tfrac{\beta}{x_1-x_0}\operatorname{ReLU}(u-x_0)+\beta),\operatorname{ReLU}(\tfrac{\beta}{x_1-x_0}\operatorname{ReLU}(l-x_0)+\beta)]+\beta \\
&= [\operatorname{ReLU}(\tfrac{\beta}{x_1-x_0}\operatorname{ReLU}(l-x_0)+\beta),\operatorname{ReLU}(\tfrac{\beta}{x_1-x_0}\operatorname{ReLU}(u-x_0)+\beta)]+\beta \\
&= [\operatorname{ReLU}(\tfrac{\beta}{x_1-x_0}\operatorname{ReLU}(l-x_0)+\beta)+\beta,\operatorname{ReLU}(\tfrac{\beta}{x_1-x_0}\operatorname{ReLU}(u-x_0)+\beta)+\beta] \\
&= [\phi_{x_0,x_1,\beta}(l),\phi_{x_0,x_1,\beta}(u)].
\end{aligned}
$$

$\square$

**Theorem 9** (Precise Monotone). Finite $\operatorname{ReLU}$ networks can IBP-express the set of monotone $\operatorname{CPWL}(\mathbb{I},\mathbb{R})$ functions precisely.

*Proof.* W.l.o.g. assume $f$ is monotonously increasing. Otherwise, consider $-f$. Let $x_i$ for $i \in \{0,\ldots,n\}$ be the set of boundary points of the linear regions of $f$ with $x_0 < \cdots < x_n$. We claim that

$$
h(x) = f(x_0) + \sum_{i=0}^{n-1} \phi_{x_i,x_{i+1},f(x_{i+1})-f(x_i)}(x)
$$

is equal to $f$ on $\mathbb{I}$ and that the IBP-analysis of $h$ is precise. We note that $f(x_{i+1}) - f(x_i) > 0$.

We first show $f = h$ on $\mathbb{I}$. For each $x \in \mathbb{I}$ pick $i \in \{1,\ldots,n\}$ such that $x_{j-1} \le x < x_j$. Then

$$
\begin{aligned}
h(x) &= f(x_0) + \sum_{i=0}^{n-1} \phi_{x_i,x_{i+1},f(x_{i+1})-f(x_i)}(x) \\
&= f(x_0) + \sum_{i=0}^{j} \phi_{x_i,x_{i+1},f(x_{i+1})-f(x_i)}(x) \\
&= f(x_0) + \sum_{i=0}^{j-1} \phi_{x_i,x_{i+1},f(x_{i+1})-f(x_i)}(x) + \phi_{x_j,x_{j+1},f(x_{j+1})-f(x_j)}(x) \\
&= f(x_0) + \sum_{i=0}^{j-1} [f(x_{i+1})-f(x_i)] + \tfrac{f(x_{j+1})-f(x_j)}{x_{j+1}-x_j}x \\
&= f(x_j) + \tfrac{f(x_{j+1})-f(x_j)}{x_{j+1}-x_j}(x-x_j) \\
&= f(x),
\end{aligned}
$$

where we used the piecewise linearity of $f$ in the last step.

Now we show that the analysis of IBP of $h$ is precise. Consider the box $[l, u] \subseteq \mathbb{I}$. We have

$$
\begin{aligned}
h([l, u]) &= f(x_0) + \sum_{i=1}^{n} \phi_{x_i, x_{i+1}, f(x_{i+1}) - f(x_i)}([l, u]) \\
&= f(x_0) + \sum_{i=1}^{n-1} [\phi_{x_i, x_{i+1}, f(x_{i+1}) - f(x_i)}(l), \phi_{x_i, x_{i+1}, f(x_{i+1}) - f(x_i)}(u)] \\
&= [f(x_0) + \sum_{i=1}^{n-1} \phi_{x_i, x_{i+1}, f(x_{i+1}) - f(x_i)}(l), f(x_0) + \sum_{i=1}^{n-1} \phi_{x_i, x_{i+1}, f(x_{i+1}) - f(x_i)}(u)] \\
&= [h(l), h(u)] \\
&= [f(l), f(u)].
\end{aligned}
$$

$\square$

## B.2 DeepPoly-0

**Lemma 10** (Convex encoding). *Let $f \in \mathrm{CPWL}(\mathbb{I}, \mathbb{R})$ be convex. Then $f$ is encoded by*

$$
h(x) = b + cx + \sum_{i=1}^{n-1} \gamma_i \, \mathrm{ReLU}(\pm_i(x - x_i)), \tag{1}
$$

*for any choice $\pm_i \in \{-1, 1\}$, if $b$ and $c$ are set appropriately, where $\alpha_i = \frac{f(x_{i+1}) - f(x_i)}{x_{i+1} - x_i}$ is the slope between points $x_i$ and $x_{i+1}$, and $\gamma_i = \alpha_i - \alpha_{i-1} > 0$ the slope change at $x_{i+1}$.*

*Proof.* First, we show that $\gamma_j = \alpha_j - \alpha_{j-1}$. Evaluating $h(x)$ for $x_j \leq x \leq x_{j+1}$ yields

$$
\begin{aligned}
h(x) &= b + cx + \sum_{i=1}^{n-1} \gamma_i \, \mathrm{ReLU}(\pm_i(x - x_i)) \\
&= b + cx + \sum_{i=1}^{n-1} \gamma_i \pm_i (x - x_i)[\pm_i = +, x_i < x] + \sum_{i=1}^{n-1} \gamma_i \pm_i (x - x_i)[\pm_i = -, x_i > x] \\
&= b + cx + \sum_{i=1}^{n-1} \gamma_i (x - x_i)[\pm_i = +, x_i < x] - \sum_{i=1}^{n-1} \gamma_i (x - x_i)[\pm_i = -, x_i > x] \\
&= b + cx + \sum_{i=1}^{j} \gamma_i (x - x_i)[\pm_i = +] - \sum_{i=j+1}^{n-1} \gamma_i (x - x_i)[\pm_i = -].
\end{aligned}
$$

The derivative of $h$ evaluated at $x$ for $x_j \leq x \leq x_{j+1}$ is $\alpha_j$:

$$
\frac{\partial h}{\partial x}(x) = c + \sum_{i=1}^{j} \gamma_i [\pm_i = +] - \sum_{i=j+1}^{n-1} \gamma_i [\pm_i = -] = \alpha_j.
$$

By choosing $\epsilon$ small enough we can ensure that $x_j + \epsilon \in [x_j, x_{j+1}]$ and $x_j - \epsilon \in [x_{j-1}, x_j]$ and thus

$$
\begin{aligned}
\alpha_j - \alpha_{j-1} &= \frac{\partial h}{\partial x}(x_j + \epsilon) - \frac{\partial h}{\partial x}(x_j - \epsilon) \\
&= \sum_{i=1}^{j} \gamma_i [\pm_i = +] - \sum_{i=j+1}^{n-1} \gamma_i [\pm_i = -] - \sum_{i=1}^{j-1} \gamma_i [\pm_i = +] + \sum_{i=j}^{n-1} \gamma_i [\pm_i = -] \\
&= \gamma_j [\pm_j = +] + \gamma_j [\pm_j = -] \\
&= \gamma_j
\end{aligned}
$$

Next, we show that one can pick $\pm_i$ arbitrarily as long as $b$ and $c$ are set appropriately. Pick any choice of $\pm_i \in \{-1, 1\}$ and set $b$ and $c$ to

$$b := f(x_0) - x_0 \frac{f(x_1) - f(x_0)}{x_1 - x_0} - \sum_{i=1}^{n-1} \gamma_i x_i [\pm_i = -] = f(x_0) - x_0 \alpha_0 - \sum_{i=1}^{n-1} \gamma_i x_i [\pm_i = -]$$

$$c := \alpha_0 + \sum_{i=1}^{n-1} \gamma_i [\pm_i = -].$$

We have $h(x) = f(x)$. Indeed: For any $x \in [x_0, x_n]$ pick $j$ s.t. $x \in [x_j, x_{j+1}]$. Then

$$h(x) = b + cx + \sum_{i=1}^{j} \gamma_i (x - x_i)[\pm_i = +] - \sum_{i=j+1}^{n-1} \gamma_i (x - x_i)[\pm_i = -]$$

$$= \underbrace{b - \sum_{i=1}^{j} \gamma_i x_i [\pm_i = +] + \sum_{i=j+1}^{n-1} \gamma_i x_i [\pm_i = -]}_{\text{offset}} + x \underbrace{\left( c + \sum_{i=1}^{j} \gamma_i [\pm_i = +] - \sum_{i=j+1}^{n-1} \gamma_i [\pm_i = -] \right)}_{\text{linear}}.$$

The offset evaluates to

$$b - \sum_{i=1}^{j} \gamma_i x_i [\pm_i = +] + \sum_{i=j+1}^{n-1} \gamma_i x_i [\pm_i = -]$$

$$= f(x_0) - x_0 \alpha_0 - \sum_{i=1}^{n-1} \gamma_i x_i [\pm_i = -] - \sum_{i=1}^{j} \gamma_i x_i [\pm_i = +] + \sum_{i=j+1}^{n-1} \gamma_i x_i [\pm_i = -]$$

$$= f(x_0) - x_0 \alpha_0 - \sum_{i=1}^{j} \gamma_i x_i [\pm_i = -] - \sum_{i=1}^{j} \gamma_i x_i [\pm_i = +]$$

$$= f(x_0) - x_0 \alpha_0 - \sum_{i=1}^{j} \gamma_i x_i$$

$$= f(x_0) - x_0 \alpha_0 - \gamma_1 x_1 - \gamma_2 x_2 - \cdots - \gamma_j x_j$$
$$= f(x_0) - x_0 \alpha_0 - (\alpha_1 - \alpha_0) x_1 - (\alpha_2 - \alpha_1) x_2 - \cdots - (\alpha_j - \alpha_{j-1}) x_j$$
$$= f(x_0) - x_0 \alpha_0 + \alpha_0 x_1 - \alpha_1 x_1 + \alpha_1 x_2 - \alpha_2 x_2 - \cdots + \alpha_{j-1} x_j - \alpha_j x_j$$
$$= f(x_0) + (x_1 - x_0) \alpha_0 + (x_2 - x_1) \alpha_1 \cdots + (x_j - x_{j-1}) \alpha_{j-1} - \alpha_j x_j$$
$$= f(x_0) + (f(x_1) - f(x_0)) + (f(x_2) - f(x_1)) \cdots + (f(x_j) - f(x_{j-1})) - \alpha_j x_j$$
$$= f(x_j) - \alpha_j x_j,$$

where we used that $\gamma_i = \alpha_i - \alpha_{i-1}$ and $\alpha_i = \frac{f(x_{i+1}) - f(x_i)}{x_{i+1} - x_i}$. The linear part evaluates to

$$c + \sum_{i=1}^{j} \gamma_i [\pm_i = +] - \sum_{i=j+1}^{n-1} \gamma_i [\pm_i = -]$$

$$= \alpha_0 + \sum_{i=1}^{n-1} \gamma_i [\pm_i = -] + \sum_{i=1}^{j} \gamma_i [\pm_i = +] - \sum_{i=j+1}^{n-1} \gamma_i [\pm_i = -]$$

$$= \alpha_0 + \sum_{i=1}^{j} \gamma_i [\pm_i = -] + \sum_{i=1}^{j} \gamma_i [\pm_i = +]$$

$$= \alpha_0 + \sum_{i=1}^{j} \gamma_i$$

$$= \alpha_0 + \sum_{i=1}^{j} \alpha_i - \alpha_{i-1}$$

$$= \alpha_i.$$

Combining the results, we get

$$h(x) = f(x_j) - \alpha_j x_j + x\alpha_j = f(x_j) + \alpha_j(x - x_j) = f(x),$$

by the piecewise linearity of $f$. □

**Lemma 25** (DP-0 Monotone ReLU). *The DP-0-analysis of the 1-layer ReLU network $h(x) = \sum_{i=1}^{n} \gamma_i \operatorname{ReLU}(x - x_i)$ yields*

$$\left.\begin{array}{l} h(x), \\ h(x_{j-1}) + \alpha_{j-1}(x - x_{j-1}) \end{array}\right\} \leq h(x) \leq \begin{cases} h(x), & \text{if all ReLUs are stable,} \\ \frac{h(u)-h(l)}{u-l}(x - l) + h(l) & \text{otherwise.} \end{cases}$$

*where $x_i \in \mathbb{R}$ s.t. $1 \leq i \leq n$ and $i < p \Rightarrow x_i < x_p$, and $\gamma_i$ are either all $> 0$ or all $< 0$. $j$ is the smallest $i$ such that $x_i \geq l$ and $k$ is the largest $i$ such that $x_i < u$. Thus, DP-0 analysis for $h(x)$ is precise.*

*Proof.* W.o.l.g. assume $h$ is monotonously increasing. Otherwise, consider $-h$.

The cases $u < x_1$ and $x_n < l$ are immediate. Choose $j$ as the smallest $i$ such that $x_i \geq l$ and $k$ as the largest $i$ such that $x_i < u$.

DP-0 yields for $\operatorname{ReLU}(x - x_i)$ on $l \leq x \leq u$

$$\left.\begin{array}{ll} x - x_i & \text{if } i \leq j, \\ 0 & \text{if } j < i < k, \\ 0 & \text{if } k \leq i, \end{array}\right\} \leq \operatorname{ReLU}(x - x_i) \leq \begin{cases} x - x_i & \text{if } i < j, \\ \frac{u-x_i}{u-l}(x - l) & \text{if } j \leq i \leq k, \\ 0 & \text{if } k < i. \end{cases}$$

Thus we have

$$\sum_{i=1}^{j-1} \gamma_i(x - x_i) \leq h(x) \leq \sum_{i=1}^{j-1} \gamma_i(x - x_i) + \sum_{i=j}^{k} \gamma_i \frac{u-x_i}{u-l}(x - l).$$

The term $\sum_{i=1}^{j-1} \gamma_i(x - x_i)$ can be simplified as follows

$$\begin{aligned}
\sum_{i=1}^{j-1} \gamma_i(x - x_i) &= \sum_{i=1}^{j-1} \gamma_i x - \sum_{i=1}^{j-1} \gamma_i x_i \\
&= x\sum_{i=1}^{j-1}(\alpha_i - \alpha_{i-1}) - \sum_{i=1}^{j-1}(\alpha_i - \alpha_{i-1})x_i \\
&= x(\alpha_{j-1} - \alpha_0) - \sum_{i=1}^{j-1}\alpha_i x_i + \sum_{i=1}^{j-1}\alpha_{i-1} x_i \\
&= x(\alpha_{j-1} - \alpha_0) - \sum_{i=1}^{j-1}\alpha_i x_i + \sum_{i=1}^{j-2}\alpha_i x_{i+1} + \alpha_0 x_1 \\
&= x(\alpha_{j-1} - \alpha_0) - \alpha_{j-1}x_{j-1} - \sum_{i=1}^{j-2}\alpha_i x_i + \sum_{i=1}^{j-2}\alpha_i x_{i+1} + \alpha_0 x_1 \\
&= x(\alpha_{j-1} - \alpha_0) - \alpha_{j-1}x_{j-1} + \sum_{i=1}^{j-2}\alpha_i(x_{i+1} - x_i) + \alpha_0 x_1 \\
&= \alpha_{j-1}(x - x_{j-1}) + \alpha_0(x_1 - x) + \sum_{i=1}^{j-2}(h(x_{i+1}) - h(x_i)) \\
&= \alpha_{j-1}(x - x_{j-1}) + \alpha_0(x_1 - x) + h(x_{j-1}) - h(x_1) \\
&= h(x_{j-1}) + \alpha_{j-1}(x - x_{j-1}),
\end{aligned}$$

hence we have proven the lower bound.

Now we consider the upper bound. We evaluate the upper bound at $l$ and $u$. If the two linear upper bounds coincide there, they coincide everywhere:

$$x = l \longrightarrow \sum_{i=1}^{j-1} \gamma_i (l - x_i) + \sum_{i=j}^{k} \gamma_i \frac{u - x_i}{u - l}(l - l)$$

$$= \sum_{i=1}^{j-1} \gamma_i (l - x_i)$$

$$= h(l) = \frac{h(u) - h(l)}{u - l}(l - l) + h(l),$$

$$x = u \longrightarrow \sum_{i=1}^{j-1} \gamma_i (u - x_i) + \sum_{i=j}^{k} \gamma_i \frac{u - x_i}{u - l}(u - l)$$

$$= \sum_{i=1}^{j-1} \gamma_i (u - x_i) + \sum_{i=j}^{k} \gamma_i (u - x_i)$$

$$= \sum_{i=1}^{k} \gamma_i (u - x_i)$$

$$= h(u) = \frac{h(u) - h(l)}{u - l}(u - l) + h(l),$$

hence we have proven the upper bound. $\qquad\square$

**Theorem 11** (DP-0 Convex). For any convex CPWL function $f \colon \mathbb{I} \to \mathbb{R}$, there exists exactly one network of the form $h(x) = b + \sum_{i \in \mathcal{I}} \gamma_i \operatorname{ReLU}(\pm_i (x - x_i))$ encoding $f$, with $|\mathcal{I}| = n - 1$ if $f$ has slope zero on some segment and otherwise $|\mathcal{I}| = n$, such that its DP-0-analysis is precise, where $\gamma_i > 0$ for all $i$.

*Proof.* The proof works as follows:

- If the function is monotonously increasing, we show that using a local argument $\pm_j = +$ and if the function is monotonously decreasing, we argue $\pm_j = -$. If $f$ has somewhere zero slope, it will be on a piecewise linear region at the boundary of $[x_0, x_n]$, in which case we need 1 neuron less.

- If the function is non-monotone and has slope zero somewhere, then there are two minima $x^*$ and $x^{**}$ that are also switching points. Hence $f$ is for all $x_j > \frac{x^* + x^{**}}{2}$ increasing and for all $x_j < \frac{x^* + x^{**}}{2}$ decreasing, so we can reuse the argument from before and need $n - 1$ neurons for that. Then we argue separately for the ReLU at $x^*$ and $x^{**}$.

- If the function is non-monotone and has nowhere slope zero, then there exists a unique minimum $x_j$. We then show that there is exactly one splitting of $\gamma_j$ to $\operatorname{ReLU}(x - x_j)$ and $\operatorname{ReLU}(-x + x_j)$.

- Finally, we prove that network is precise.

We know that there are finitely many switching points $x_i$, $1 \le i \le n$.

Case 1: $f$ is monotone. W.o.l.g. assume $f$ is monotonously increasing; the proof is similar for monotonously decreasing $f$. Assume that $\pm_j = -$ for some $j$. Then there exists $\epsilon > 0$ such that for all $x \in [x_j - \epsilon, x_j + \epsilon]$, all the ReLUs except $\operatorname{ReLU}(-x + x_j)$ are stable, i.e., either always 0 or always active. Further, for such inputs, DP-0 yields

$$0 \le \gamma_i \operatorname{ReLU}(x - x_j) \le \frac{\gamma_j}{2}(-x + x_j + \epsilon).$$

As $f$ has no minimum, at least one other ReLU need to active at $x_j$: If there would be no other active ReLU, then $f$ would have a minimum. The active neuron(s) contributes a linear term of the form $\beta x$, $\beta \ne 0$, hence we get for $x_j - \epsilon \le x \le x_j + \epsilon$ and some $b \in \mathbb{R}$, DP-0 bounds are

$$b + \beta x \le h(x) \le b + \beta x + \tfrac{1}{2}(-x + x_j + \epsilon).$$

We know that for $x = x_j$, we get $h(x_j)$ and thus $h(x_j) = b + \beta x_j$. The slope of $h$ at $x_j - \epsilon$ is $\beta - \gamma_j$. Hence, we have $h(x_j - \epsilon) = h(x_j) - (\beta - \gamma_j)\epsilon > b + \beta(x_j - \epsilon) = h(x_j) - \beta\epsilon$ since $\gamma_j \epsilon > 0$. This contradicts the assumption that we are precise since the lower bound from DP-0 analysis $h(x_j) - \beta$ is unequal to the actual lower bound $h(x_j - \epsilon)$. Hence, we have to have $\pm_j = +$ for all $j$.

Case 2: $f$ is not monotone and has slope 0 somewhere. Then, there are two minima which are also switching points, $x^*$ and $x^{**}$. The argument in Case 1 is a local one, hence we can use the same argument for all $i$ such that $x_i \notin \{x^*, x^{**}\}$, so we only need to consider the cases for $x_j$ and $x_{j+1}$ such that $x_j = x^*$ and $x_{j+1} = x^{**}$. Since $f$ is convex, the only possible case is that $f$ is monotonously decreasing for $x < x_j$ and monotonously increasing for $x > x_{j+1}$, thus we have $\pm_k = -$ for $k < j$ and $\pm_k = +$ for $k > j + 1$. Now we claim that $\pm_j = -$ and $\pm_{j+1} = +$: We have either $(\pm_j, \pm_{j+1}) = (-, +)$ or $(\pm_j, \pm_{j+1}) = (+, -)$. If not, we would not get a unique minimum. The second case also leads directly to a contradiction: Not only would the pre-factors of the two ReLU need to coincide, i.e., $\gamma_j = \gamma_{j+1}$ (otherwise one would not have the minimum between them), the analysis of $h$ on $x_j - \epsilon \le x \le x_j + \epsilon$ yields (as only the neurons at $x_j$ and $x_{j+1}$ are active there)

$$0 \le R(x - x_j) \le \tfrac{1}{2}(x - x_j + \epsilon),$$
$$-x + x_{j+1} = R(-x + x_{j+1})$$
$$b + \gamma_{j+1}(-x + x_{j+1}) \le h(x) \le b + \gamma_{j+1}(-x + x_{j+1}) + \tfrac{\gamma_{j+1}}{2}(x - x_j + \epsilon)$$

Here the lower bound is $b - \gamma_{j+1}\epsilon < b$, thus imprecise.

Case 3: $f$ is not monotone and has slope 0 nowhere. Then, there is one minimum $x^* = x_j$. For all $x_i \ne x_j$ we can argue as before. So we just need to argue about $x_j$. Assume we only have one ReLU involving $x_j$. The only unstable ReLU leads to DP-0 lower bound 0, while all others together lead to a linear term $ax + b$ for some $a \ne 0$, thus the overall lower bound from DP-0 is $h(x_j) - |a|\epsilon < h(x_j)$. Therefore, such network cannot be precise under DP-0 analysis. As one ReLU is not enough, we can try two at $x_j$, namely $\gamma'_j \operatorname{ReLU}(x - x_j)$ and $\gamma''_j \operatorname{ReLU}(-x + x_j)$. As around $\gamma_j$ no other ReLU is active, it immediately follows that we have $\gamma'_j = \alpha_j$ and $\gamma''_j = \alpha_{j-1}$.

Now we finally prove that the network constructed above is precise. Consider the input $l \le x \le u$. In the case where $f$ is monotone on $[l, u]$, we have the result immediately by using Lemma 25 as all ReLU with an opposite orientation are inactive. In the case where $f$ is on $[l, u]$ not monotone, the above construction yields

$$h(x) = b + \sum_{i, \pm_i = -} \gamma_i \operatorname{ReLU}(-(x - x_i)) + \sum_{i, \pm_i = +} \gamma_i \operatorname{ReLU}(x - x_i).$$

We can apply Lemma 25 to $\sum_{i, \pm_i = -} \gamma_i \operatorname{ReLU}(-(x - x_i))$ and $\sum_{i, \pm_i = +} \gamma_i \operatorname{ReLU}(x - x_i)$ individually to get

$$0 \le \sum_{i, \pm_i = -} \gamma_i \operatorname{ReLU}(-(x - x_i)) \le \tfrac{h(l) - b}{u - l}(-x + u),$$
$$0 \le \sum_{i, \pm_i = +} \gamma_i \operatorname{ReLU}(x - x_i) \le \tfrac{h(u) - b}{u - l}(x - l).$$

Hence the combined bounds are

$$b \le h(x) \le h(l) + \tfrac{h(u) - h(l)}{u - l}(x - l).$$

Evaluating the upper bounds at $x = l$ and $x = u$ yields $h(l)$ and $h(u)$ respectively, hence the bounds are precise. $\qquad\square$

### B.3 DEEPPOLY-1

**Corollary 12** (DP-1 ReLU). The ReLU network $h(x) = x + \operatorname{ReLU}(-x)$ encodes the function $f(x) = \operatorname{ReLU}(x)$ and, the DP-1-analysis of $h(x)$ is identical to the DP-0-analysis of ReLU. Further, the DP-0-analysis of $h(x)$ is identical to the DP-1-analysis of ReLU.

*Proof.* We first prove that $x + \operatorname{ReLU}(-x) = \operatorname{ReLU}(x)$.

$$x + \operatorname{ReLU}(-x) = \operatorname{ReLU}(x) - \operatorname{ReLU}(-x) + \operatorname{ReLU}(-x) = \operatorname{ReLU}(x).$$

Next we show that the DP-0-analysis of $R(x)$ coincides with the DP-1-analysis of $x + \operatorname{ReLU}(-x)$.

- Case $l \leq 0 \leq u$, $x \in [l, u]$: We have for DP-0 and $\mathrm{ReLU}(x)$:

$$0 \leq \mathrm{ReLU}(x) \leq \tfrac{u}{u-l}(x - l).$$

For DP-1 and $x + \mathrm{ReLU}(-x)$ we have:

$$-x \leq \mathrm{ReLU}(-x) \leq \tfrac{l}{u-l}(x - u),$$

Hence

$$0 \leq x + \mathrm{ReLU}(-x) \leq \tfrac{l}{u-l}(x - u) + x = \tfrac{u}{u-l}(x - l).$$

- Case $0 \leq l \leq u$, $x \in [l, u]$: We have for DP-0 and $\mathrm{ReLU}(x)$:

$$\mathrm{ReLU}(x) = x.$$

For DP-1 and $x + \mathrm{ReLU}(-x)$ we have:

$$\mathrm{ReLU}(-x) = 0,$$

Hence

$$x + \mathrm{ReLU}(-x) = x.$$

- Case $l \leq u \leq 0$, $x \in [l, u]$: We have for DP-0 and $\mathrm{ReLU}(x)$:

$$\mathrm{ReLU}(x) = 0.$$

For DP-1 and $x + \mathrm{ReLU}(-x)$ we have:

$$\mathrm{ReLU}(-x) = -x,$$

Hence

$$x + -x = 0.$$

To show the opposite, we only need to prove for the case $l \leq 0 \leq u$ since DP-0 and DP-1 are identical when there is no unstable ReLU. For DP-1 and $x + \mathrm{ReLU}(-x)$ we have:

$$0 \leq \mathrm{ReLU}(-x) \leq \tfrac{l}{u-l}(x - u),$$

Hence

$$x \leq x + \mathrm{ReLU}(-x) \leq \tfrac{l}{u-l}(x - u) + x = \tfrac{u}{u-l}(x - l).$$

$\square$

**Corollary 13** (DP-1 Approximation)**.** Finite ReLU networks can DP-1- and DP-0-express the same function class precisely. In particular, they can DP-1-express the set of convex functions $f \in \mathrm{CPWL}(\mathbb{I}, \mathbb{R})$ and monotone functions $f \in \mathrm{CPWL}(\mathbb{I}, \mathbb{R})$ precisely.

*Proof.* The follows immediately with Corollary 12 and the technique presented in Theorem 11. $\square$

### B.4 TRIANGLE

**Theorem 15** ($\Delta$ Convex)**.** Let $f \in \mathrm{CPWL}(\mathbb{I}, \mathbb{R})$ be convex. Then, for any network $h$ encoding $f$ as in Lemma 10, we have that its $\Delta$-analysis is precise. In particular, $\pm_i$ can be chosen freely.

*Proof.* The $\Delta$ analysis of a ReLU over some input range $l \leq x \leq u$ results in the convex hull of ReLU on that range. With that, we can apply the Lemma 14 over the network $h$ from Lemma 10 modeling $f$:

$$h(x) = b + cx + \sum_{i=1}^{n-1} \gamma_i \, \mathrm{ReLU}(\pm_i(x - x_i))$$

This is regardless of the choice of $\pm_i$ a sum of convex functions ($\gamma_i > 0$). With Lemma 14 we get that the $\Delta$ analysis of $h$ results in the convex hull of $h$, and thus is precise. $\square$

## B.5 MULTI-NEURON-RELAXATIONS

**Theorem 26** (Multi-Neuron Precision). For every $f \in \mathrm{CPWL}(\mathbb{I}, \mathbb{R})$, there exists a single layer ReLU network $h$ encoding $f$, such that its multi-neuron analysis (considering all ReLUs jointly) is precise.

*Proof.* As the multi-neuron relaxation yields the exact convex hull of all considered neurons in their input-output space, it remains to show that every $f \in \mathrm{CPWL}(\mathbb{I}, \mathbb{R})$ can be represented using a single-layer ReLU network.

Recall that every $f \in \mathrm{CPWL}(\mathbb{I}, \mathbb{R})$ can be defined by the points $\{(x_i, y_i)\}_i$ s.t. $x_i > x_{i-1}$ (Definition 1). We set $h_1(x) = (x - x_0)\frac{y_1 - y_0}{x_1 - x_0} + y_0$ and now update it as follows:

$$h_{i+1}(x) = h_i(x) + \left( \frac{y_{i+1} - y_i}{x_{i+1} - x_i} - \frac{y_i - y_{i-1}}{x_i - x_{i-1}} \right) \mathrm{ReLU}(x - x_i)$$

We observe that $\mathrm{ReLU}(x - x_i) = 0$ for all $x_j$ such that $j \leq i$. As $h(x)$ is CPWL, it is now sufficient to show that $h(x_i) = y_i, \forall i$.

$$
\begin{aligned}
h(x_j) &= (x - x_0)\frac{y_1 - y_0}{x_1 - x_0} + y_0 + \sum_{i=1} \left( \frac{y_{i+1} - y_i}{x_{i+1} - x_i} - \frac{y_i - y_{i-1}}{x_i - x_{i-1}} \right) \mathrm{ReLU}(x - x_i) \\
&= (x_j - x_0)\frac{y_1 - y_0}{x_1 - x_0} + y_0 + x_j \sum_{i=1}^{j-1} \left( \frac{y_{i+1} - y_i}{x_{i+1} - x_i} - \frac{y_i - y_{i-1}}{x_i - x_{i-1}} \right) - \sum_{i=1}^{j-1} \left( \frac{y_{i+1} - y_i}{x_i - x_{i-1}} - \frac{y_i - y_{i-1}}{x_i - x_{i-1}} \right) x_i \\
&= (x_j - x_0)\frac{y_1 - y_0}{x_1 - x_0} + y_0 + x_j \left( \frac{y_j - y_{j-1}}{x_j - x_{j-1}} - \frac{y_1 - y_0}{x_1 - x_0} \right) - x_j \frac{y_j - y_{j-1}}{x_j - x_{j-1}} \\
&\quad + \sum_{i=1}^{j-1} \frac{y_{i+1} - y_i}{x_{i+1} - x_i}(x_{i+1} - x_i) + x_1 \frac{y_1 - y_0}{x_1 - x_0} + \sum_{i=1}^{j-1} \frac{y_i - y_{i-1}}{x_i - x_{i-1}}(x_i - x_i) \\
&= y_j
\end{aligned}
$$

$\square$

