# OpenReview forum: "Expressivity of ReLU-Networks under Convex Relaxations"
_ICLR.cc/2024/Conference — ICLR 2024 poster_

### Official Review · Reviewer_pT5c · 2023-10-19

**Soundness:** 3 good
**Presentation:** 3 good
**Contribution:** 2 fair
**Rating:** 6
**Confidence:** 3

**Summary:**

This work studies the expressive power of ReLU neural networks under various convex relaxations, by measuring their ability to represent (certain subclass of) CPWL functions. On the positive side, for univariate CPWL functions, most convex relaxation methods are shown to be able to express monotone or convex CPWL functions, and Multi-Neuron can even represent all CPWL functions. However, it's shown that all these methods fail in the multivariate case: they can't even represent the simple $\max$ function in $\mathbb{R}^2$.

**Strengths:**

**Novel direction:** the expressive power of neural networks is an interesting and important topic. This work is the first to consider such expressive power in the precise representation setting.

**Wide coverage:** in the univariate setting, this work discusses a broad range of convex relaxation methods, which covers most popular ones in practice.

**Weaknesses:**

**Univariate is restricted:** the main weakness I found is the restricted univariate assumption. All the positive results for the expressive power of convex relaxation methods hold for univariate functions, which is restricted in two ways: (1) in practice, almost all functions of interest are multivariate (2) in theory, the case of univariate functions is too special, which often times avoids the general difficulty in high dimensions and thereby hard to generalize.

On the other hand, the negative multivariate result only holds for $\triangle$, and the precision gap can be arbitrarily small. It's not clear whether multivariate methods can express multivariate CPWL functions precisely.

**Questions:**

**Motivation for precise analysis:** it's mentioned in this work if we allow approximate analysis, for any approximation error $\epsilon>0$ and general multivariate continuous function on $\mathbb{R}^n$, IBP can express the function up to $\epsilon$ error (Baader et al 20). This seems a strong enough guarantee. What's the significance of considering precise analysis beyond pure theoretical interest?

---

> ### Author Response · Authors · 2023-11-14
> **Reply to Reviewer pT5c Part I**
>
> We thank the reviewer for their high-quality review and the interesting questions they raised. We are happy to hear that they find our approach novel and research question both important and interesting. Below we address their remaining questions.
>
> **Q1: Can you discuss the relevance of your results in the heavily restricted univariate domain? What are the implications of your negative results for the $\Delta$-relaxation in the multivariate setting?**
>
> First, we want to clarify that the $\Delta$-relaxation is strictly more precise than the DeepPoly and IBP relaxations. Thus, our negative result for the $\Delta$-relaxation in the multivariate setting directly implies negative results for the other, less precise relaxations (DeepPoly-0, DeepPoly-1, and IBP), even for extremely simple functions. We believe this result to be of great importance as the majority of neural network verifiers are upper bounded in their analysis precision by the so-called “convex relaxation barrier”, realized by the $\Delta$-relaxation [1]. Only a few, very recent methods use multi-neuron-relaxations (at great computational cost) to break this barrier [2,3]. This highlights that even extremely simple functions can not be encoded as ReLU networks such that these simpler verifiers can analyze them precisely and agrees well with the fact that the latter methods have recently dominated the international neural network verification competition [4].
>
> However, as even simple multivariate functions can not be encoded such that their analysis is precise using any single neuron domain $D$ ($D$ is IBP, DeepPoly-0, DeepPoly-1 or $\Delta$), we can not separate them with respect to their expressivity in this setting. To do so and investigate if the more complex domains improve expressivity, we study the univariate case, and indeed show positive results and separation between methods, both in terms of expressivity and size of resulting solution spaces.
>
> **Q2: Why do you not discuss multi-neuron relaxations in the multi-variate setting?**
> We agree with the reviewer that this is an interesting question and have given it some thought, but remain unable to prove either a positive or negative result. We note that the first (to the best of our knowledge) constructive proof by He et al (2020) [9] showing that arbitrary CPWL functions can be encoded by (finite) ReLU networks was only published recently, highlighting the difficulty of this question even in the non-robust setting. Unfortunately, a multi-neuron analysis of the networks constructed in this proof is not precise, thus an altogether different construction would be required which would be out of scope for this already dense work. To show a negative result with our technique, we would require a network simplification strategy (and counterexample) for multi-neuron relaxations. However, simple functions with one non-linearity (such as the max function) can be analyzed precisely using multi-neuron relaxations and more complex functions/subnetworks with two or more non-linearities can not generally be simplified to fewer layers, necessitating a completely different approach. Thus we believe this to be out of the scope of this work.
>
> Further, to provide some context to the contributions made in this work, we want to again highlight that prior work [6] considered only a single relaxation method (the simple IBP), while we consider four relaxations, the univariate and multivariate setting, and four function classes.
>
> **References:**
> [1] Salman, et al. "A convex relaxation barrier to tight robustness verification of neural networks." NeurIPS 2019
> [2] Ferrari et al. "Complete verification via multi-neuron relaxation guided branch-and-bound." ICLR 2022
> [3] Zhang et al. "General cutting planes for bound-propagation-based neural network verification." NeurIPS 2022
> [4] Müller et al. "The third international verification of neural networks competition (VNN-COMP 2022): summary and results." arXiv 2022
> [6] Mirman et al. “The fundamental limits of neural networks for interval certified robustness”, TMLR 2022
> [7] Baader et al. “Universal approximation with certified networks”, ICLR 2020
> [8] Wang et al. “Interval universal approximation for neural networks”, POPL 2022
> [9] He, et al. "ReLU deep neural networks and linear finite elements." JCM 2020
> [10] Müller et al. "Certified training: Small boxes are all you need."ICLR 2023
> [11] Mao et al. "Connecting Certified and Adversarial Training." NeurIPS 2023
> [12] De Palma et al. "Expressive Losses for Verified Robustness via Convex Combinations." arXiv 2023

---

> > ### Author Response · Authors · 2023-11-14
> > **Reply to Reviewer pT5c Part II**
> >
> > **Q3: What's the significance of your results on exactly analyzable networks beyond pure theoretical interest, given that Baader et al. [7] show that IBP is sufficient if we tolerate approximation errors of $\epsilon > 0$?**
> > First, we believe it is of inherent interest to understand what function class certified networks can express. While standard (non-certified) ReLU networks, can express all CPWL functions (continuous piecewise linear functions) [9], our work shows that convex relaxations are unexpectedly restrictive: not even very simple multivariate functions that only require a single ReLU in the standard (non-certified) setting can be certifiably expressed by any finite ReLU network. In fact, while constructive proofs in prior work [7, 8] require network widths to go to infinity as the approximation error $\epsilon$ goes to 0, our work shows that such infinite width is not only sufficient but necessary to express interesting (multivariate) functions.
> > Second, despite the universal approximation results for IBP [7,8] and great interest from the community [10,11,12], training networks to good certifiably robust accuracy even on simple datasets (e.g. Cifar-10) remains an open problem. To us, this indicates that approximation up to an $\epsilon$ error is insufficient for good empirical performance. We thus believe that investigating the limits of current (certified training) methods that are based on convex relaxations is essential for significant progress on this challenging task. We believe, that our work constitutes an important step in this direction and provides new perspectives on the certified training problem that might prove to be of great relevance to the field.
> > We thank the reviewer for raising this question and are happy to highlight these points in the paper.
> > We hope to have been able to address the reviewer's concerns regarding both the motivation for and scope of our work, are happy to answer any follow-up questions, and are looking forward to their reply.
> >
> > **References:**
> > [1] Salman, et al. "A convex relaxation barrier to tight robustness verification of neural networks." NeurIPS 2019
> > [2] Ferrari et al. "Complete verification via multi-neuron relaxation guided branch-and-bound." ICLR 2022
> > [3] Zhang et al. "General cutting planes for bound-propagation-based neural network verification." NeurIPS 2022
> > [4] Müller et al. "The third international verification of neural networks competition (VNN-COMP 2022): summary and results." arXiv 2022
> > [6] Mirman et al. “The fundamental limits of neural networks for interval certified robustness”, TMLR 2022
> > [7] Baader et al. “Universal approximation with certified networks”, ICLR 2020
> > [8] Wang et al. “Interval universal approximation for neural networks”, POPL 2022
> > [9] He, et al. "ReLU deep neural networks and linear finite elements." JCM 2020
> > [10] Müller et al. "Certified training: Small boxes are all you need."ICLR 2023
> > [11] Mao et al. "Connecting Certified and Adversarial Training." NeurIPS 2023
> > [12] De Palma et al. "Expressive Losses for Verified Robustness via Convex Combinations." arXiv 2023

---

> > ### Comment · Reviewer_pT5c · 2023-11-22
> >
> > Thanks for the reply. Although I still think univariate is a big restriction, given the lower bounds in the multivariate setting and the fact that this work is the first result of the kind, considering univariate is forgivable and I have raised the score accordingly.

---

> > > ### Author Response · Authors · 2023-11-22
> > > **Thanks for engaging in the discussion and follow-up**
> > >
> > > We thank the reviewer for engaging in the discussion and are happy to hear that they appreciate our results.
> > >
> > > Given the reviewer's focus on our positive result in the univariate setting, we want to highlight that we do consider the **multivariate** case and show that even the most restricted function class of monotone and convex function **can not be expressed** precisely using **any single-neuron convex relaxation** (including IBP, DP-0, DP-1, and $\Delta$).
> > >
> > > While our results in the univariate setting allow us to show clear advantages of more precise relaxations, we believe the negative result in the multivariate setting is particularly interesting and relevant as it implies that expressing any non-trivial function precisely under convex relaxation requires more precise relaxations than are typically used in the field.

---

### Official Review · Reviewer_y6LY · 2023-10-27

**Soundness:** 2 fair
**Presentation:** 3 good
**Contribution:** 2 fair
**Rating:** 5
**Confidence:** 4

**Summary:**

This paper studies the expressive power of ReLU neural networks under different convex relaxations that are commonly used for neural network certification. The key findings are:

* For univariate functions, more precise convex relaxations like Δ and DeepPoly allow expressing larger classes of continuous piecewise linear (CPWL) functions precisely compared to the simple interval bound propagation (IBP).
* IBP can precisely express monotone CPWL functions, while Δ and DeepPoly can also express convex CPWL functions.
* Multi-neuron relaxations allow single-layer networks to express all univariate CPWL functions precisely.
* For multivariate functions, even the most precise single-neuron relaxation (Δ) cannot precisely express simple classes like multivariate monotone convex CPWL functions.
* This suggests single-neuron convex relaxations are fundamentally limited for multivariate functions, highlighting the need for more precise analysis methods like multi-neuron relaxations.
* The results have implications for certified training, suggesting more precise relaxations could yield larger effective hypothesis spaces and higher performance if optimization challenges can be overcome.

In summary, the paper provides an in-depth analysis of the expressive power of ReLU networks under different convex relaxations, showing more precise relaxations increase expressivity for univariate functions but are still fundamentally limited for multivariate functions. The results motivate developing more advanced analysis techniques and studying their potential benefits for certified training.

**Strengths:**

Here are some strengths of this paper:

* It provides the first in-depth, systematic study on the expressive power of ReLU networks under a wide range of convex relaxations commonly used in neural network certification.
* The analysis covers univariate and multivariate functions and simple as well as more complex function classes like (monotone) convex CPWL.
* It clearly differentiates the capabilities of different relaxations through precise mathematical results and constructive proofs.
* The paper relates the theoretical results back to certified training, drawing interesting hypotheses about the potential benefits of more advanced relaxations.
* The paper is clearly structured and provides detailed mathematical proofs for all results.

**Weaknesses:**

Some potential weaknesses of this paper:

* The focus is exclusively on ReLU networks, not covering other activation functions commonly used like sigmoid or tanh.
* Only fully-connected feedforward networks are considered, not convolutional or residual architectures widely used in practice.
* The analysis is limited to deterministic networks, not touching on stochastic networks.
* Only standardized datasets and perturbation sets are studied; results may not generalize to other domains.
* While hypotheses are provided for certified training, no experiments are conducted to validate the conjectured benefits.
* The writing is quite dense and mathematical, which could make it less accessible to a general AI audience.
* Aside from certified training, implications for other applications of neural network analysis are not discussed much.

**Questions:**

Here are my questions:

* There is recent literature on the convex optimization of ReLU network, e.g., [1] and several other follow-up papers by the same authors extending this work to various neural network architectures. Can authors comment on their contributions over this work and explain how their paper supports/refutes the claim there? For example, the very first question to comment on would be: What is the point of having convex relaxations if the ReLU networks can already be trained using convex optimization?

* Can authors also briefly comment on the issues raised in the weaknesses part?

[1] Pilanci and Ergen, Neural Networks are Convex Regularizers: Exact Polynomial-time Convex Optimization Formulations for Two-layer Networks, ICML 2020

---

> ### Author Response · Authors · 2023-11-14
> **Reply to Reviewer y6LY Part I**
>
> We thank the reviewer for their review, high-quality summary, and great outline of the strengths of our paper. However, we have identified some misconceptions and factual errors in the listed weaknesses which we address below while answering the remaining questions.
>
> **Q1: Can you discuss why you focus exclusively on ReLU networks instead of considering sigmoid or tanh activations?**
> We decided to focus on ReLU networks for two key reasons.
> Firstly, they are much more widespread and thus relevant in the certified robustness community, e.g., the international verification of neural networks competition last year considered ReLU activations in all 12/12 benchmark classes, sigmoids in only one, and tanh in none at all [1].
> Secondly, finite sigmoid and tanh networks can not exactly encode (to the best of our knowledge) any commonly used function class, making the question of exact analysis less relevant and interesting. Given the limited space and already “dense and mathematical” writing of our work, we decided to focus on the most relevant class of networks.
>
> **Q2: You seem to only consider fully connected feedforward networks. Can you discuss the extensibility of your results to other architectures?**
> We already consider arbitrary ReLU networks with any architecture including arbitrary skip connections, which include both CNNs and ResNets (see the first paragraph in Section 2 and Theorem 17). We have made this more clear.
>
> **Q3: Why do you only consider deterministic neural networks and not stochastic networks?**
> As our work is motivated by (deterministic) neural network verification (where all state-of-the-art methods are based on convex relaxations), we focus on the general type of network typically considered in the field, i.e., deterministic networks. In particular, we are not aware of any such method considering stochastic networks. Perhaps the reviewer can point to a specific class of stochastic networks that would be relevant in this setting.
>
> **Q4: You only seem to study standardized datasets and perturbation sets. Can you discuss to what extent your results generalize beyond that?**
> As we do not study any specific data(set) anywhere in the paper but argue about general function classes, our results will generalize to any applicable dataset.
> For univariate functions, we show a positive result for the general case of continuous bounded perturbation sets, subsuming all $\ell_p$-norm bounded perturbations, typically studied in the field. In the multivariate case, we show a negative result even for the more restrictive (but very commonly used) $\ell_\infty$-norm bounded perturbations. That is, even extremely simple functions can not be expressed for box input regions. This entails that these functions can also not be expressed for more general classes of perturbation sets. We note that Theorem 19 and thus 20 can be trivially extended to other $\ell_p$-norms and have added a corresponding Theorem (Thm. 24) to Appendix A.
>
> **Q5: While hypotheses are provided for certified training, no experiments are conducted to validate the conjectured benefits.**
> First, we want to highlight that our general results agree well with multiple recent empirical works as discussed in “Implications of our Results for Certified Training” (Section 1).
> While validating our conjecture of more precise relaxations being beneficial for certified training is an interesting item for future work, it is beyond the scope of this theoretical work, as no current certified training method supports more precise relaxations, as current approaches are non-differentiable.
>
> **Q6: The writing seems to be quite dense and mathematical - how accessible is it to a general AI audience?**
> As work on certified adversarial robustness [2,3,4], in general, and closely related prior theoretical work [5,6], specifically, is frequently published at general AI venues (including ICLR), we believe our work to also be accessible to the audience of these venues, given that multiple reviewers highlighted the good presentation.
>
> **References:**
> [1] Müller et al. "The third international verification of neural networks competition (VNN-COMP 2022): summary and results." arXiv 2022
> [2] Xu et al. “Fast and Complete: Enabling Complete Neural Network Verification with Rapid and Massively Parallel Incomplete Verifiers” ICLR 2021
> [3] Ferrari et al. "Complete verification via multi-neuron relaxation guided branch-and-bound." ICLR 2022
> [4] Müller et al. “Certified training: Small boxes are all you need” ICLR 2023
> [5] Baader et al. “Universal approximation with certified networks”, ICLR 2020
> [6] Mirman et al. “The fundamental limits of neural networks for interval certified robustness”, TMLR 2022
> [7] Pilanci and Ergen, Neural Networks are Convex Regularizers: Exact Polynomial-time Convex Optimization Formulations for Two-layer Networks, ICML 2020

---

> > ### Author Response · Authors · 2023-11-14
> > **Reply to Reviewer y6LY Part II**
> >
> > **Q7: Aside from certified training, can you discuss the implications of your results for other applications of neural network analysis?**
> > Generally, convex relaxations of neural networks are most frequently used in certified training and neural network certification, which are both very active research areas. Beyond the implications for certified training, which we discuss in detail in Section 1, our results directly imply the need for precise certification methods to obtain exact analysis results even for trivial multi-variate functions.
> >
> > **Q8: How does your work relate to Pilanci and Ergen’s [1] results on training ReLU networks using convex optimization?**
> > Pilanci and Ergen [7] tackle a completely different problem in a different setting. While they consider standard training, we consider the existence of certifiably adversarial robust networks. While they cast training of neural networks as a convex optimization problem in the network parameters, we consider convexity with respect to the network inputs, which is generally unrelated. As we discuss the existence of ReLU networks with certain properties, irrespective of how they might be trained, we believe their result on the training problem to be completely unrelated to our results. Further, their approach, based on analyzing the dual of the optimization problem, is unrelated to ours, which is based on bound relaxations in primal space. Finally, they consider fitting finite datasets, while we consider the expression of function classes. If the reviewer could point us to any results of Pilanci and Ergen’s [3] that address any of the questions we investigate, we are more than happy to discuss how they relate to ours.
> >
> > We hope to have been able to address the misconceptions perceived as potential weaknesses of this paper as well as the reviewer’s question, are happy to answer any follow-up points and look forward to the reviewer's response.
> >
> > **References:**
> > [1] Müller et al. "The third international verification of neural networks competition (VNN-COMP 2022): summary and results." arXiv 2022
> > [2] Xu et al. “Fast and Complete: Enabling Complete Neural Network Verification with Rapid and Massively Parallel Incomplete Verifiers” ICLR 2021
> > [3] Ferrari et al. "Complete verification via multi-neuron relaxation guided branch-and-bound." ICLR 2022
> > [4] Müller et al. “Certified training: Small boxes are all you need” ICLR 2023
> > [5] Baader et al. “Universal approximation with certified networks”, ICLR 2020
> > [6] Mirman et al. “The fundamental limits of neural networks for interval certified robustness”, TMLR 2022
> > [7] Pilanci and Ergen, Neural Networks are Convex Regularizers: Exact Polynomial-time Convex Optimization Formulations for Two-layer Networks, ICML 2020

---

> ### Comment · Reviewer_y6LY · 2023-11-21
> **Response to rebuttal**
>
> I'd like to thank the authors for their detailed responses. The rebuttal clarified some of my concerns therefore I increase my score.

---

> > ### Author Response · Authors · 2023-11-21
> > **Thank you for the reply and follow up**
> >
> > We thank the Reviewer for engaging in the discussion and are glad to hear that we could address some of their concerns.
> >
> > We would be delighted if the Reviewer could mention which of their concerns remain undressed and will try our best to clarify these points as well.

---

### Official Review · Reviewer_FMX3 · 2023-11-09

**Soundness:** 2 fair
**Presentation:** 1 poor
**Contribution:** 2 fair
**Rating:** 8
**Confidence:** 2

**Summary:**

The authors presented an analysis for special classes of convex relaxations of neural networks and their expressivity. Unfortunately, given that I am not an expert with the subject, and that the background in the paper is limited, I have trouble understanding the high level ideas of the paper. I would encourage the authors to engage with discussion, so that I can provide a proper review of this work.

**Strengths:**

I will re-evaluate the strengths after a discussion with the authors to understand the paper.

**Weaknesses:**

Similar to strengths, I will re-evaluate after discussion.

**Questions:**

As mentioned earlier, I have many basic questions about this work. Perhaps let me start from the basics about the background of this work.

1. Can the authors elaborate on how convex relaxations are helping with formal robustness guarantees? This was quickly glossed over, but I would hope to get a better explanation.

2. Can the authors explain exactly what is being relaxed into a convex function? Section 2.1 reads quite confusingly for me, as all I can discern are inequalities, and I do not see which parts are being relaxed.

3. When the authors write a vector is less than or equal to another vector, is this inequality entrywise?

4. There are a lot of definitions in section 2.2, can the authors provide a simpler explanation of these definitions and what they are trying to capture? In particular, why is it that we care about $D$-analysis in the definition of expressivity? I would have expected expressivity of a network architecture to be about function approximation.

5. At the end, it seems like the authors are investigating whether or not ReLU networks can express certain functions. I thought this was already an answered question with respect to universal approximation, but perhaps I am missing something. Can the authors explain why we need to analyze the expressivity results for these certain class of functions?

Perhaps let's start here. Once we go into discussion and have a better understanding, I can follow up with more questions regarding the actual technical contributions of this work.

---

> ### Author Response · Authors · 2023-11-14
> **Reply to Reviewer FMX3**
>
> We thank the reviewer for their candid admission that they are not an expert in the field and are happy to provide a significantly extended version of the background, which we believe addresses all the reviewer's questions. We have incorporated some of these extensions (including a better intuition for Definition 3) into Section 2 in an effort to make our work more accessible.
>
> ## Extended Background
> **Problem Setting**
> For clarity of exposition, we focus on the most common case of adversarially robust classification.
> We call a classifier $h: \mathcal{X} \to \mathcal{Y}$ locally robust around some input $x$ if it predicts the same, correct class for all similar inputs $\mathcal{B}^\epsilon_p(x) := \\{ x' \in \mathcal{X} \mid \| x - x' \|_p \leq \epsilon \\}$. Thus, to prove that a classifier is locally robust, we have to show  $\forall x' \in \mathcal{B},  h(x') = h(x) = y$.
> For a neural network that predicts the class $ h(x) \:= argmax_i f(x)_i $, we can equivalently show that the logit of the target class is always greater than that of all other classes, by solving the following optimization problem:
>
> $0 < \min_{x' \in \mathcal{B},  i \neq y}  f(x')_y - f(x')_i $
>
> **Convex Relaxations for Efficient Solution**
> Unfortunately, solving this problem exactly is NP-complete [1] as the neural network $f$ is highly non-linear and non-convex. Thus, state-of-the-art neural network verifiers use convex relaxations (combined with a branch-and-bound approach), to relax the above non-convex optimization problem to a (convex) linearly constrained linear program which can be solved much more efficiently [2]. More concretely, the non-linear activation layers (ReLUs), which can be seen as non-linear constraints in the original problem, are replaced with a set of linear constraints (the element-wise inequalities) in their input-output space. Intuitively, we relax the non-convex graph of the ReLU function with the convex over-approximations illustrated in Figures 1-4. However, as these convex-relaxations introduce an (over)-approximation error, these relaxed problems might be too imprecise to prove robustness even when the network is robust. In this work, we study for which classes of functions there exist ReLU networks such that this analysis does not lose precision depending on the relaxation method used.
>
> **Relevance of a $D$-Analysis**
> Solving the above optimization problem can geometrically be interpreted as linearly projecting the graph of the network (set of input-output tuples) for inputs in $\mathcal{B}$ to the 1d-line $f_y - f_i$ corresponding to the output difference for every $i \neq y$. While this projection is computationally efficient, computing the exact graph is still NP-complete. Thus, we can instead use convex relaxations to compute and then project (often in one step) an over-approximation of this graph and then check whether all points in this over-approximation are classified correctly. We call this over-approximate graph the $D$-analysis depending on the convex relaxation $D$ that was used. As the analysis with relaxation $D$ is precise if and only if the 1d projection of the exact graph and the $D$-analysis agree, this is a very natural and intuitive perspective on the problem.
>
> We can now reformulate our above question as “For which function classes exist ReLU networks such that the 1d projection of their $D$-analysis (over-approximation of the graph) is precise, i.e., agrees with the projection of the exact graph?”. For “expressivity under convex relaxation” we thus do not only require that a given function can be encoded by a network (the default notion of expressivity and already shown for CPWL functions and ReLU networks by prior work [3]) but additionally require the analysis of the resulting network using the $D$ relaxation to be precise. Thus, $D$-expressivity depends on both the relaxation $D$ and the function class $F$.
>
> For a more in-depth explanation of (certified) adversarial robustness including examples and visualizations, we recommend the excellent blog post by Eric Wong: https://locuslab.github.io/2019-03-12-provable/
>
> **Notation**
> All inequalities between vectors are element-wise. We have added a corresponding note to the Notation paragraph. Thanks for the pointer.
>
> We are looking forward to the reviewer’s reply and are happy to provide further explanations or references and answer any follow-up questions.
>
> **References:**
> [1] Katz et al. "Reluplex: An efficient SMT solver for verifying deep neural networks." CAV 2017
> [2] Salman, et al. "A convex relaxation barrier to tight robustness verification of neural networks." NeurIPS 2019
> [3] He, et al. "ReLU deep neural networks and linear finite elements." JCM 2020

---

> > ### Comment · Reviewer_FMX3 · 2023-11-15
> > **Response**
> >
> > I want to start by thanking the authors for the detailed reply and the updated draft. I believe the context is a lot clearer to me now, and I think I understand the problem of interest. At this point, I do think the results are interesting, but I would like clarify a few more things, with the current work.
> >
> > Firstly, I want to understand the negative result in two dimensions a bit better. Is the issue of expressivity a problem with the convex relaxation, or is it something more basic about ReLU networks? What I mean is, can f(x,y) = max(x,y) be expressed by a finite sized ReLU network at all?
> >
> > If the answer is no, then in some sense, the convex relaxation of this problem isn't very interesting, since in this case the definition of "precise" is clearly too strong. I think a better approach would be studying the $\epsilon$-precise version of this expressivity problem.
> >
> > Secondly, I want to understand the impact on training actual networks in practice better. If I was a practitioner who only cares about training adversarially robust networks, can the authors explain how I might have to modify my training methods given a potential future result from this line of work?

---

> > > ### Author Response · Authors · 2023-11-16
> > > **Reply to Follow-Up of Reviewer FMX3**
> > >
> > > We thank the reviewer for their prompt reply and are happy to hear that we were able to address all their questions and that they now believe our results to be interesting. Regarding the follow-up questions:
> > >
> > > **Q1: Can ReLU networks express the function $f(x,y) = \max(x,y)$ precisely?**
> > > Yes, ReLU networks can express (we call this ``encode’’ in the non-robust setting as per Definition 2) the $\max$ function.
> > >
> > > In fact, He et al. [1] show that all (multivariate) continuous piecewise linear (CPWL) functions and thus all function classes we consider (including the $max$ function) can be expressed precisely by ReLU networks (first paragraph of the related work, Section 3).
> > >
> > > The max function can even be expressed precisely using just a single neuron (see the proof to Theorem 19 in Appendix A):
> > > Consider the network $ h(x,y) = y + \text{ReLU}(x-y) $, then $h$ is equal to $\max$ for all $x, y \in \mathbb{R}$:
> > > * Consider the case $x \geq y$, then $\text{ReLU}(x-y) = x-y$ as $x - y \geq 0$. Hence $h(x,y) = y + \text{ReLU}(x-y) = y + x - y = x = \max(x,y)$.
> > > * Consider the case $x < y$, then $\text{ReLU}(x - y) = 0$ as $x - y < 0$. Hence $h(x,y) = y + \text{ReLU}(x-y) = y + 0 = y = \max(x,y)$
> > > * Hence, we have $h(x,y) = \max(x,y)$.
> > >
> > > While other constructions are possible as well, we first show that the above yields the most precise $\Delta$-abstraction of *all* ReLU networks (Theorem 17 and 18), and still it can not precisely $\Delta$-express the max function (Theorem 19).
> > >
> > > To summarize, we only consider function classes that can be encoded precisely by ReLU networks and then show that, in the multivariate setting, even under severe restrictions (monotonicity and convexity of the function) they can not be $\Delta$-expressed, i.e., their analysis with the most precise single-neuron relaxation is imprecise for *any* ReLU network. Thus, we show that this is a limitation of convex relaxations.
> > >
> > > **Q2: What are the takeaways for a practitioner aiming to train a (certifiably) adversarially robust network?**
> > > Great question: Current certified training methods are based on the imprecise IBP relaxation [2,3] as the more precise relaxations lead to harder optimization problems that current optimization strategies can not solve efficiently [4]. So far, it was an open question whether these more precise relaxations would yield any advantages if we could address the optimization issues. Our results now suggest that (at least in the univariate case), more precise relaxations allow larger function classes to be $D$-expressed precisely and lead to larger solution spaces for the same function class. Thus, our results directly suggest that working on these optimization issues is a promising path forward to improve certified training and obtain networks with higher certified robust accuracy.
> > > We provide a high-level discussion of this question in the last paragraph of the first Section ``Implications of our Results for Certified Training’’.
> > >
> > > **References**
> > > [1] He, et al. "ReLU deep neural networks and linear finite elements." JCM 2020
> > > [2] Müller et al. Certified Training: Small Boxes are All You Need, ICLR 2023
> > > [3] Shi et al. Fast Certified Robust Training with Short Warmup, NeurIPS 2021

---

> > > > ### Comment · Reviewer_FMX3 · 2023-11-20
> > > > **Response**
> > > >
> > > > Thanks for the detailed reply. I think I understand the problem and contributions much better now. I will raise my score to accept.
> > > >
> > > > On the other hand, while I cannot make the other reviewers engage and raise their scores, I hope the discussion has helped the authors improve the draft.

---

> > > > > ### Author Response · Authors · 2023-11-21
> > > > > **Thank you for Updating your Review**
> > > > >
> > > > > We thank the reviewer for engaging actively in the discussion and updating their overall score as a result. We would like to ask them to consider reevaluating their the partial scores on Soundness, Presentation, and Contribution in light of our rebuttal and paper revision.

---

### Author Response · Authors · 2023-11-19
**Rebuttal Reminder**

As the rebuttal period is slowly drawing to a close, we would like to encourage the Reviewers to pose any follow-up questions they might have and let us know if we were unable to address all of their concerns.

In any case, we would greatly appreciate the Reviewers taking the time to acknowledge our rebuttal.

---

### Meta-Review · Area_Chair_dV5P · 2023-12-14

**Metareview:**

Authors study the expressive power of finite ReLU neural networks by considering their various convex relaxations. They measure networks ability to represent continuous piecewise linear (CPWL) functions.

In the univariate case, most convex relaxation methods succeed. In the multivariate case, it's shown that these methods fail. I think the paper is overall interesting and should be included in ICLR. My impression from reviewer comments is that paper needs some clarification; the authors should carefully go over and clarify all reviewers' questions/concerns.

This paper was reviewed by 3 reviewers and received the following Rating/Confidence scores: 8/2, 6/3, 5/4. The reviewer who is championing the paper has a rather low confidence score, and increased their score after rebuttal. Subsequent AC/reviewer discussions led to acceptance.

**Justification For Why Not Higher Score:**

It is a good paper but not award worthy.

**Justification For Why Not Lower Score:**

It is a good paper, so it should be included in the program.

---

### Decision · Program_Chairs · 2024-01-16

Accept (poster)